# Natural Products and Altered Metabolism in Cancer: Therapeutic Targets and Mechanisms of Action

**DOI:** 10.3390/ijms25179593

**Published:** 2024-09-04

**Authors:** Wamidh H. Talib, Media Mohammad Baban, Mais Fuad Bulbul, Esraa Al-Zaidaneen, Aya Allan, Eiman Wasef Al-Rousan, Rahaf Hamed Yousef Ahmad, Heba K. Alshaeri, Moudi M. Alasmari, Douglas Law

**Affiliations:** 1Faculty of Allied Medical Sciences, Applied Science Private University, Amman 11931, Jordan; 2Faculty of Health and Life Sciences, Inti International University, Nilai 71800, Negeri Sembilan, Malaysia; douglas.law@newinti.edu.my; 3Department of Clinical Pharmacy and Therapeutics, Applied Science Private University, Amman 11931, Jordan; mediababan22@gmail.com (M.M.B.); maisbulbul95@gmail.com (M.F.B.); i.alzayadeen@ju.edu.jo (E.A.-Z.); ayaallan92@yahoo.com (A.A.); eiman_alrousan@hotmail.com (E.W.A.-R.); rahafhamed77799@outlook.com (R.H.Y.A.); 4Department of Pharmacology, Faculty of Medicine, King Abdul-Aziz University, Rabigh 25724, Saudi Arabia; halshaeri@kau.edu.sa; 5College of Medicine, King Saud bin Abdulaziz University for Health Sciences (KSAU-HS), Jeddah 21423, Saudi Arabia; asmarim@ksau-hs.edu.sa; 6King Abdullah International Medical Research Centre (KAIMRC), Jeddah 22233, Saudi Arabia

**Keywords:** natural products, Warburg effect, anticancer, angiogenesis, metastasis, human health

## Abstract

Cancer is characterized by uncontrolled cell proliferation and the dysregulation of numerous biological functions, including metabolism. Because of the potential implications of targeted therapies, the metabolic alterations seen in cancer cells, such as the Warburg effect and disruptions in lipid and amino acid metabolism, have gained attention in cancer research. In this review, we delve into recent research examining the influence of natural products on altered cancer metabolism. Natural products were selected based on their ability to target cancer’s altered metabolism. We identified the targets and explored the mechanisms of action of these natural products in influencing cellular energetics. Studies discussed in this review provide a solid ground for researchers to consider natural products in cancer treatment alone and in combination with conventional anticancer therapies.

## 1. Introduction

Many metabolic pathways are reprogrammed by oncogenes in most tumors, and this reprogramming is necessary for tumor initiation and essential to the tumor’s continued growth and multiplication [1]. Cancer cells independently alter their flux through various metabolic pathways to meet the increased demand for bioenergetic and biosynthetic resources as well as to lower oxidative stress, which, in turn, is essential for cancer cell proliferation and survival. Moreover, cancer driver mutations regulate the flux of these metabolic pathways in conjunction with the availability of nutrients in the surrounding environment. Furthermore, when metabolites build up abnormally, they can also aid in developing tumors [2]. In addition, compared to their nontumor counterparts, tumor cells typically metabolize glucose, lactate, pyruvate, hydroxybutyrate, acetate, glutamine, and fatty acids at significantly higher rates [3]. The discovery of the mitochondrial respiratory chain complex IV by Otto Warburg, who was awarded the 1931 Nobel Prize in Medicine or Physiology, is credited with helping to establish the basis of cancer metabolism [4]. Warburg noted that cancer tissue slices often utilize copious quantities of glucose to produce lactate, even in the presence of oxygen, in contrast to normal tissues. This phenomenon is known as aerobic glycolysis or the Warburg effect. According to Warburg, an “injury to respiration” caused by cancer cells is necessary for a differentiated cell to change into a proliferative cancer cell [5]. Moreover, tumor cells have been shown to have altered interactions involving various intermediates of glycolysis, the TCA cycle, the ETC, the pentose phosphate pathway, and lipid metabolism pathways, all of which have been implicated in the development of tumors [6].

As the field of oncology advances, there is a growing recognition of the potential to modulate these metabolic pathways for therapeutic benefits.

According to the literature, numerous natural products are available as chemoprotective agents against common cancers that occur globally. It has long been known that natural products contain a wide variety of bioactive substances. Many of these substances have anti-inflammatory, antioxidant, and anti-proliferative qualities, which in turn make them desirable candidates for cancer biology interventions. These natural products can be found in vegetables, fruits, plant extracts, and herbs [7,8]. In addition, natural products can treat tumors safely and effectively. By optimizing their structure, they can also serve specific biological functions [9]. Moreover, by targeting the glycolytic/metabolic phenotype, natural products can inhibit the process of glycolysis and impede tumor growth and migration. Increasingly, the signaling pathways and important components involved in aerobic glycolysis are regulated by natural products. Three pathways exist for natural products to control aerobic glycolysis. Firstly, natural products influence glycolysis in tumor cells by controlling glycolytic enzymes directly. Secondly, natural products have the potential to regulate genes associated with aerobic glycolysis by modulating oncogenes such as HIF-1α, MYC, and p53, which could alter a tumor cell’s metabolic pathways. Thirdly, natural products have the ability to inhibit the glycolysis of tumor cells by acting on the PI3K/Akt/mTOR or AMPK pathways [10]. For instance, the polyphenol resveratrol, which is present in grapes, prevents glycolysis by triggering AMP-activated protein kinase (AMPK), which in turn prevents colon cancer from spreading and invading [11]. More recently, research has attempted to clarify how natural products might affect cancer cells’ altered metabolism, opening up new therapeutic research directions. As cancer is characterized by metabolic reprogramming, natural products have been identified as possible cellular metabolism modulators with implications for cancer therapy and prevention. Hence, this review will focus on recent articles so as to investigate the impact of natural products on altered metabolism in cancer.

## 2. Cellular and Cancer Metabolism

Cells require energy for survival, growth, and division, which is obtained through absorbing nutrients like glucose. This energy is broken down through metabolic processes such as glycolysis and OXPHOS-mediated cellular respiration. Normal cells primarily rely on OXPHOS for energy production, but cancer cells proliferate at a rate more than the angiogenesis capacity, leading to low oxygen levels. Glycolysis occurs in cancer cells even when oxygen is present, leading to increased glycolytic ATP production [12,13,14,15]. The Warburg effect of aerobic glycolysis has been identified as the leading cause of elevated glycolysis in cancer cells. However, research has shown that most cancer cells do not exhibit mitochondrial damage. However, aerobic glycolysis can occur concurrently to improve energy production for the long-term maintenance of cancer cell homeostasis [15,16]. Tumor cells rely on glutamine as a metabolic substrate and energy source, which impacts amino acid transport, autophagy, and metabolism [17] (Figure 1). Metabolic reprogramming in cancer cells is regulated by multiple pathways, including PI3K/Akt, which enhances glucose uptake and glycolysis [18,19]. AMPK regulates nutrient availability, promotes tumorigenesis under metabolic stress, and activates mTORC1/2, which controls the downstream signaling of metabolism, apoptosis, protein and lipid synthesis, and cell survival [20,21,22]. Oncogenes like c-Myc and HIF-1α are master inducers of cancer glycolysis via the direct or indirect transactivation of cancer glycolytic genes [23]. C-Myc expression in cancer cells promotes energy production and anabolic processes, leading to rapid proliferation without growth factor stimulation [24]. In addition, the main cause of increased glycolysis and lactate production during hypoxia is HIF-1α [25]. In normal O2 conditions, HIF-1α upregulates the genes for glycolytic enzymes, erythropoietin (EPO), glucose transporter 1 (GLUT1), and vascular endothelial growth factor (VEGF) [26,27]. In addition to increasing the rate of glycolysis, HIF-1α activation suppresses oxidative phosphorylation by upregulating genes like LDH-A and pyruvate dehydrogenase kinase 1 (PDK1) (Figure 1). Pyruvate’s entry into the TCA cycle is inhibited [28]. It has been suggested that HIF-1α inhibition could be a therapeutic approach to treat cancer because of the role of HIF-1α in aerobic glycolysis and mitochondrial function [27,28,29,30]. Figure 1 represents the main metabolic pathways.

## 3. Natural Products Targeting Cancer Metabolism

### 3.1. Resveratrol (RV)

Resveratrol (RV) is the most compelling polyphenolic compound. It is present in a wide range of fruits and vegetables, including peanuts, grapes, and sprouts. RV was first isolated from *Veratrum grandiflorum*, also known as white hellebore plants [31]. Resveratrol’s chemical structure consists primarily of two aromatic rings linked by a methylene bridge. It occurs naturally in *trans*- and *cis*-isomeric forms. It has been noted that *trans*-resveratrol and its glucoside provide a variety of benefits, such as anti-tumor, cardioprotective, anti-oxidative, and anti-inflammatory qualities [32]. Resveratrol is a highly effective scavenger of radicals produced by cells and triggered by metals, enzymes, hydroxyls, and superoxides. It also guards against reactive oxygen species (ROS)-induced DNA damage and lipid peroxidation within cell membranes [33]. Preclinical research has extensively examined the potential use of resveratrol as a nutraceutical and therapeutic agent for a wide range of disorders. Cancer patients are particularly interested in using it because of the substantial risks involved with traditional therapies like chemotherapy and surgery [34]. The effects of resveratrol on tumor initiation, development, and progression were initially observed in 1997. Since then, numerous studies have shown its extensive prophylactic and curative benefits against a wide range of cancer types, including liver tumors, gastrointestinal tract, breast, lung, and prostate cancers. When combined with other cytostatic medications, resveratrol exhibits notable chemo-preventive effects on malignancies by targeting numerous in vitro and in vivo pathways, further highlighting its therapeutic potential [35].

It is known that malignant cells are highly dependent on the glycolytic pathway to meet their energy and metabolic intermediate needs when it comes to cancer. Even under aerobic conditions, they switch from oxidative phosphorylation to glucose fermentation as their primary ATP-producing process [16]. These cells have a greater need for glucose than normal cells because of this metabolic shift, which produces less ATP per glucose molecule. Cancer cells frequently overexpress the glucose transporter 1 (GLUT1), a facilitative carrier to maintain a steady source of glucose [36,37]. Targeting glucose metabolism as a therapeutic method in cancer treatment has a biological basis in the differences in energy metabolism between normal and cancerous cells [38]. Inhibiting glucose transporters in neoplastic cells is a promising cancer treatment method that seeks to create an energy-deprivation condition that can enhance the effects of other anticancer medicines [39]. One important way resveratrol affects cancer metabolism is by interfering with signaling pathways essential for cell division and survival. As mentioned above, malignant cells supply their demand for glucose by overexpressing transporters such as GLUT1 [36,37]. Resveratrol has been demonstrated to inhibit the cell membrane transport by GLUT1 via the Akt/mTOR-dependent signaling pathway [40]. The inhibition of the Akt/mTOR-dependent signaling pathway reduces anabolic pathways, decreases glucose uptake, and produces less lactate in a variety of cancer cell lines via down-regulating PKM2 expression [41]. Kueck et al.’s study, which evaluated the impact of RV on the viability and glucose uptake of five human ovarian cancer cell lines, revealed that treatments lasting up to eight hours could decrease cell viability, lactate production, glucose uptake, Akt, and mammalian target of rapamycin (mTOR) signaling in a manner that was dependent on time and dose [42,43]. The authors concluded that autophagy-mediated cell death was triggered in ovarian cancer cells by the energy deprivation state generated by RV [44].

The metabolism of cancer cells requires the high absorption of glucose; this process is tightly controlled and involves several components, including growth factors [45]. Within healthy cells, intracellular glucose is phosphorylated by hexokinase to produce glucose-6-phosphate, which is then transformed into 3-carbon pyruvate, producing ATP and NADH. Rather than fermenting glucose, healthy cells use oxidative phosphorylation to efficiently synthesize ATP and CO_2_ when oxygen is present (a process known as aerobic glycolysis) [46]. On the other hand, tumor cells rely heavily on the fermentation, or breakdown, of glucose [47]. Regardless of oxygen supply, most cancer cells display an altered metabolism defined by increased glycolysis and lactate generation; this is called the Warburg effect [5]. Resveratrol has been shown to modify the Warburg effect, as shown in Figure 2 [48,49].

According to the authors of [50], who demonstrated that the tumor cells are weakened, effective anti-tumor therapy is supported by a low pH tumor microenvironment, which is linked to several aspects such as reduced vascularization, food deprivation, and hypoxia in the setting of the Warburg effect.

Tumor cells need a steady and consistent supply of nucleotides for DNA synthesis and energy to fuel their hyperactive proliferation. The pentose phosphate pathway (PPP) provides these essential commodities. Remarkably, many PPP enzymes exhibit severe dysregulation within malignancies. The initial step in the PPP is catalyzed by glucose-6-phosphate-dehydrogenase (G6PDH), an essential enzyme that controls the rate at which tumor cells grow. Elevated G6PDH expression and PPP activity are common in malignant tumors [51,52,53,54]. Since G6PDH knockdown dramatically decreases cell growth, targeting G6PDH with targeted inhibition may be a helpful therapy option for glycolytic malignancies [55]. For survival, tumor cells transition to aerobic glycolysis. Consequently, the primary enzyme for identifying the glycolytic phenotype of tumor cells is lactate dehydrogenase (LDH), which catalyzes the conversion of pyruvate to lactate. As a result, LDH may be used as a therapeutic target. Indeed, the advancement of lymphomas and pancreatic cancer xenografts is inhibited by LDH inhibition [56]. There are more glycolytic intermediates available for biosynthetic pathways, including PPP, which speeds up the growth of tumors and the synthesis of macromolecules like ribose-5-phosphate (R5P) [57]. R5P is a crucial pathway of pentose phosphate intermediate that acts as a precursor to the synthesis of macromolecules [58]. The procedures that turn food carbon into FAs are part of lipid synthesis. Glycerol-3-phosphate transforms FAs into diacylglycerides and TAGs, which create the glycerol backbones of the lipids [59,60]. FAs are esterified to phospholipids (PLs) in cancer cells to facilitate membrane lipid production and encourage cell division [61]. Research has demonstrated that apoptosis is induced when FASN is blocked in cancer cells, and cell growth arrest occurs [62,63]. RV dramatically decreased lipid production in many cancer cell lines by downregulating FASN [64,65]. Several genes related to FA and cholesterol production are induced to express themselves through the phosphoinositide-3-kinase/Akt/mammalian target of rapamycin (PI3K/Akt/mTOR) pathway [66,67]. Sterols, which are lipids other than cholesterol and cholesteryl-esters, are also essential for the operation of membranes [68,69]. The structural foundation for the production of steroid hormones, including progesterone and estrogen, is provided by cholesterol [60,68,69]. The production of FA and cholesterol involves a class of proteins called sterol regulatory element-binding proteins (SREBPs) [60]. SREBPs, mediated by 3-hydroxy-3-methyl glutaryl coenzyme A reductase (HMGCR), may cause abnormally high cholesterol levels [70]. RV blocked the mevalonate pathway in rat theca-interstitial cells, lowered HMGCR expression and activity, and decreased cholesterol production [71]. Additionally, AMP-regulated protein kinase (AMPK) targets SREBPs [72]. RV action is shown in multiple mechanisms by Professor Ido’s group. Initially, AMPK was activated by RV-induced SIRT1 activation [73]. They further postulated that the unpredictability of this cascade could cause inconsistent RV effects. They discovered that RV’s action as a SIRT1 activator might come from an integrated SIRT1-liver kinase B1 (LKB1)–AMPK impact in addition to SIRT1 activation. Thus, RV activates SIRT1 by directly binding to SIRT1 and boosting nicotinamide adenine dinucleotide (NAD)+ levels by upregulating the salvage pathway through nicotinamide phosphoribosyl transferase (NAMPT) activation, an effect mediated by AMPK. In addition to encouraging the deacetylation of a select few SIRT1 substrate proteins (such as PGC-1α, peroxisome proliferator-activated receptor gamma coactivator 1-alpha), it also stimulates other sirtuins besides SIRT1. If cellular energy generation is already compromised in the course of cancer treatment, RV may expedite cell death [74]. High doses of RV (4 g/kg body weight/day) have been demonstrated to activate AMPK in a way that is SIRT1-independent, indicating that dosage is a crucial component of RV functionality. Consequently, it is possible that RV-induced SIRT1 activation contributes to AMPK activation in vivo and in vitro [75]. Sirtuins are NAD-dependent enzymes that are highly conserved. The mammalian sirtuin family impacts numerous cellular functions, such as DNA repair, lipid and glucose metabolism, and carcinogenesis [76]. SREBP1, a hepatic transcription factor for lipogenesis and cholesterol production, is deacetylated and destabilized by SIRT1 by focusing on the genes involved in cholesterol production; SREBP2 regulates cholesterol homeostasis [77,78]. Maintaining cholesterol homeostasis involves liver-X-receptor (LXR) proteins, which sense cholesterol. By inducing the expression of the ATP-binding cassette transporter A1, LXRs improve the reverse transport of cholesterol from peripheral tissues (ABCA1). High-density lipoproteins are formed when these transport proteins transfer cholesterol to apolipoprotein AI. Lipoproteins are soluble lipid–protein complexes created when apolipoproteins bind with lipids. The three main lipids that circulate in plasma are cholesterol, TAGs, and PLs [77].

### 3.2. Curcumin (CUR)

Curcumin ((1E,6E)-1,7-bis(4-hydroxy-3-methoxyphenyl)-1,6-heptadiene-3,5-dione) is isolated from the dried root of *Curcuma longa*, a member of the ginger family, Zingeberaceae [79]. Approximately 80% curcumin, 18% demethoxycurcumin, and 2% bis-demethoxycurcumin are present in commercial-grade turmeric. It is fully soluble in ethanol or dimethyl sulfoxide but only partially soluble in water. Steam distillation has been used to separate essential oils from the turmeric root, including a-phellandrene (1%), sabinene (0.6%), cineol (1%), borneol (0.5%), zingiberene (25%), and sesquiterpenes (53%) [80]. Most recently, studies have shown that curcumin is a powerful anti-inflammatory and antioxidant substance with a variety of medicinal uses [81]. Curcumin regulates cytokines, different kinase proteins, growth factors, and their receptors, all of which are involved in the growth and advancement of the cell cycle [82]. Traditional Indian and Chinese medicine both employed curcumin as a therapy for various illnesses.

Curcumin has been used for medical purposes since the Unani and Vedic eras. In addition, it obstructs the transmission mechanisms and molecular targets connected to the start and development of different malignancies. Curcumin has been proven to be an effective treatment for several types of cancer, as shown in Figure 3.

It has been shown that by independently blocking the PI3K/Akt/mTOR pathway, plumbagin and curcumin both have anticancer action. Their combined impact on this route is still unknown. A growing body of research indicates that the combination of medications shows better efficacy in treating cancer [83,84]. Curcumin has been shown to increase radiation-induced apoptosis in human Burkitt’s lymphoma cells by blocking the PI3K/Akt pathway and its downstream NF-κB protein expression [85].

### 3.3. Thymoquinone (TQ)

Thymoquinone, an active component obtained from *Nigella sativa* seeds, is used extensively in treating various illnesses due to its documented antibacterial, antioxidant, anticancer, and anti-inflammatory properties. In various cancer types, such as bladder, colon, pancreatic, neuroblastoma, osteosarcoma, myeloblastic leukemia, and acute lymphoblastic leukemia, thymoquinone demonstrates cytotoxic effects by preventing cell proliferation and triggering cell apoptosis (Figure 4).

Thymoquinone altered the Bax, Bcl2, and cytochrome c protein levels and caused mitochondrial dysfunction in T24 and 253J bladder cancer cells. The apoptosis-promoting effect of thymoquinone may be partially reversed by pretreatment with the pan-caspase inhibitor Z-VADfmk. This effect is achieved by promoting the expression of the antiapoptotic protein Bcl2, inhibiting the translocation of Bax from the cytoplasm to mitochondria, and preventing the release of cytochrome c [86]. In the irinotecan-resistant (CPT-11-R) LoVo colon cancer cell line, thymoquinone was also discovered to stimulate autophagic cell death and trigger mitochondrial outer membrane permeability [87]. Thymoquinone, betulinic acid, and gemcitabine pretreatments, together with gemcitabine, effectively inhibited the growth of cancer cells in vitro by downregulating the expression of PKM2, a promising part of cellular metabolism [88]. Thymoquinone caused mitochondrial apoptosis in acute lymphocyte leukemic CEM-ss cells by activating caspases 3 and 8 and producing cellular ROS [89].

### 3.4. Allicin

Poor Man’s Treacle is a member of the Amaryllidaceae family, which includes garlic (*Allium sativum* L.). Garlic bulbs are composed of 65% water, 28% carbohydrates, 2% organosulfur compounds, 2% proteins (mostly allinase), 1.2% amino acids, and 1.5% fiber [90].

Throughout history, many civilizations have used garlic in cooking, but its most widespread application has been in Mediterranean cuisine, specifically in the Middle East and Asia. It is believed to have been planted in the Middle East around 5000 years ago and is among the oldest plants still in cultivation, along with other growing crops. Moreover, garlic has been used historically and currently as a medicinal for many years [91].

Prevention of damage to DNA and anti-inflammatory properties: finding natural substances that can stop or limit DNA damage is essential because it is a necessary step at the beginning of the entire carcinogenic process. Allicin can increase indirect DNA protection (antioxidant activity and oxidizing enzyme regulation), direct DNA protection, and immunity regulation.

Reactive oxygen species (ROS) and reactive nitrogen species (RNS) are highly reactive chemicals. Because they take part in many signaling mechanisms, cellular metabolism frequently produces ROS and RNS. Oxidative and nitrosative stress happens when the quantity of ROS and RNS exceeds the antioxidant systems’ capacity. Subsequently, ROS and RNS impair cellular functionality by causing damage to cell components, including DNA. Because of this, squelching free radicals is a tactic to stop the formation of tumors, as they are involved in cancer development [92].

Few studies were conducted to verify the in vivo anticancer potential of allicin. Allicin was tested on a variety of tumor mouse models, including cholangiocarcinoma [93], colon cancer [94], and lymphoma [95]. These studies showed the potential of allicin to inhibit tumor growth in vivo. Regarding the in vitro tests, the precise mechanism of action remains to be determined. Allicin was able to modify STAT3 and decrease the tumor burden in colorectal tumor models [94] and cholangiocarcinoma [93]. Apoptosis is a sign of allicin’s antagonistic action on tumor growth in mice harboring lymphomas [95]. Similarly, intratumor injection of a low dose of allicin (500 µg/mouse every other day for a sequential 14 days) inhibited tumor growth in mice xenografted with hepatocellular carcinoma cells, causing both intrinsic and extrinsic apoptosis, more so than the positive control DOX (20 µg/mouse), partially elevating Bax and FASL mRNA [96]. The combination of 10 µg of allicin and 20 µg of DOX produced the greatest results for both tumor development and apoptotic marker expression [96]. This indicates that allicin can be utilized as an adjuvant chemical. Figure 5 shows various mechanisms that allicin utilizes to fight cancer.

### 3.5. Genistein

Genistein is a small biologically active isoflavonoid found in high concentrations in soybeans. It is a natural phytoestrogen that is associated with a variety of beneficial activities such as antioxidant, anti-inflammatory, antibacterial, antiviral, cholesterol regulation, osteoporosis prevention, and anticancer effect [97,98]. The mechanisms of genistein to inhibit cancer are comprehensive and involve a considerable number of molecular pathways [99].

Genistein has a structural similarity with natural estrogen, allowing it to block estrogen’s hormonal activity, which makes it a perfect candidate to manage estrogen-dependent cancers like breast cancer, ovarian cancer, and endometrial (uterine) cancer [98]; likewise, decreasing the symptoms of postmenopausal [100]. Genistein has been proven preclinically effective against various types of human cancers such as breast, lung, liver, prostate, pancreatic, skin, cervical, bone, uterine, colon, kidney, bladder, neuroblastoma, gastric, esophageal, pituitary, salivary gland, testicular, and ovarian cancers [98].

Genistein exhibits its activity through two approaches: firstly, the chemoprevention by inhibiting the formation of COX-2 and oxidative stress; secondly, by targeting carcinogenesis possible pathways and interfering with their mechanism via suppressing cancer cell proliferation, metastasis, invasion, tumor angiogenesis, regulation of epigenetic, activation of survival pathway, matrix metalloproteinase (MMP), and vascular endothelial growth factor (VEGF) [98].

One way is by inhibiting cancer metastasis with dual activity exerting effects on the initial steps of primary tumor growth and the later steps of the metastatic cascade [97]. It works by inhibiting the phosphorylation of Focal Adhesion Kinase (FAK) that is responsible for activating the signaling pathway to control cell migration [100].

Additionally, genistein can inhibit the vascular endothelial growth factor (VEGF) by lowering the activity of cyclooxygenase-2 (COX-2) responsible, as well as matrix metalloproteinase 9 (MMP-9) expression, which is highly expressed in cancer cells, either by activating the AMPK pathway or by deactivating the PIP3 pathway [99].

Genistein is an anti-tumoricidal molecule in different types of cancer (Figure 6); it can induce apoptosis by mediating various signaling cascades, including caspases through extrinsic and intrinsic mitochondrial pathways [101]. 

Another mechanism for apoptosis induction is simply by interfering with Akt (also known as serine-threonine protein kinase B), which is needed for the reaming of cell survival. Genistein inhibits Akt/NF-κB pathways, inducing apoptosis as seen in breast and prostate cancer [100].

Genistein downregulates hypoxia-inducible factor-1α (HIF-1α), therefore inactivating glucose transporter 1 (GLUT1) or/and hexokinase 2 (HK2), which in turn suppresses aerobic glycolysis and mediates apoptosis [100].

Genistein has proven its ability to restrain the formation of new blood vessels by inhibiting the expression of vascular endothelial growth factor (VEGF), which is a key regulator of angiogenesis. It also downregulates platelet-derived growth factor, urokinase plasminogen activator, matrix metalloprotease-2 (MMP-2), and MMP-9 expression, as seen in bladder cancer and oral squamous cell carcinoma [101]. Another way to explain the effects of genistein anticancer activity on cancer cells is by deregulating the cell cycle to ensure the regulation of cell growth and cell cycle progression in cancer cells, modulating the expression of cell cycle-regulatory proteins to arrest the cycle in different phases. Genistein has been found to arrest the cell cycle progression at the G2-M phase in human gastric carcinoma cells as well as in galectin-3-transfected human breast epithelial cell [101]. Although there is significant evidence about the potential preventive and therapeutic effects against cancer, a significant consideration regarding its safety at high doses must be considered [102]. 

## 4. Epigallocatechin Gallate

Globally, millions of individuals consume green tea daily, and its polyphenol compounds have been widely studied for their anticancer properties. Among these, epigallocatechin gallate (EGCG), a major catechin in green tea, has shown the most potent antiproliferative effects. In a comparative study of ten polyphenols, including caffeic acid (CA), gallic acid (GA), catechin (C), epicatechin (EC), and their derivatives, EGCG exhibited a solid ability to induce apoptosis and cause G1 phase cell cycle arrest in cancer cells [103].

While GA also demonstrated some antiproliferative effects, the esterification of C and EC with GA to form CG and ECG (epicatechin gallate) significantly enhanced these properties [103,104].

A comparable correlation was discovered between EGCG and EGC. The gallic acid group significantly increased the anticancer properties of catechins. This characteristic might be used to synthesize flavonoid derivatives to create brand-new anticancer drugs [104].

The synergistic effects of EGCG with other compounds, such as ginseng derivatives, further underscore its role in cancer chemoprevention. For instance, combining EGCG with panaxadiol (PD), a purified component of ginseng, has been shown to significantly inhibit colon cancer cell proliferation [103,105]. Previous studies observed that ROS levels were decreased by combining therapy with ginseng and antioxidants. ROS accumulated in ginseng-treated colorectal cancer cells activated a cellular signaling defense system. The results of a pilot investigation indicated that epicatechin, but not catechin, the two naturally occurring antioxidants in green tea, increased the anticancer activity of panaxadiol (PD), a purified ginseng component. A further study found that using PD combined with epigallocatechin gallate (EGCG), a key catechin found in green tea, synergistically affected colon cancer cells [103].

The most prevalent and potent antioxidant in green tea for the chemoprevention of cancer is EGCG. Green tea may work well with anticancer medications to prevent cancer by increasing their inhibitory effects on colon cancer cell proliferation, as demonstrated by an earlier study where EGCG amplified the effects of ginseng components in this regard. Despite these findings, a comprehensive comparison of the distinct effects of various tea polyphenols on colon cancer prevention remains to be conducted. Figure 7 illustrates the proposed mechanisms by which EGCG may inhibit key cancer hallmarks.

## 5. Piperine

Piperine (1-Piperoylpiperidine) is an alkaloid component found in fruits, black pepper (*Piper nigrum*), or long pepper (*Piper longum*). It is a daily consumed dietary phytochemical all around the world due to the popularity of the black pepper as a spice; it is the reason for its distinguished biting taste. Plus, it is responsible for the range of benefits of pepper because of its anti-inflammatory, immunosuppressive, and antibiotic characteristics which makes it a great candidate for many therapeutic applications [106,107].

It was tested on different types of cancer cells, and it showed great therapeutic potential for breast cancer, ovarian cancer, gastric cancer, glioma cancer, lung cancer, oral squamous cell carcinoma, prostate cancer, colorectal cancer, cervical cancer, leukemia [108], melanoma, fibrosarcoma, and osteosarcoma [107].

Piperine has a diverse mechanism of action, which gives it two unique properties as a chemopreventive agent: it can block the initiation of tumors and suppress the transformation of initiated cells into neoplastic cells [109].

In addition to its prevention action, it can potentially be considered a therapeutic agent that can target and tackle cancer cells without harming normal cells, overcoming the biggest obstacle in classical pharmacological treatments of cancer [109].

Piperine interferes with the abnormal mechanisms of the cancer cells in many ways. One approach is to induce the death of cells. Piperine can induce various cell death types, including apoptosis and autophagy, as seen in breast and prostate cancer. Piperine can trigger the intrinsic apoptotic pathway by releasing mitochondrial cytochrome c, activating caspase-3 and -9, cleaving poly-ADP ribose polymerase (PARP), and inactivating p38/MAPK (mitogen-activated protein kinase) and NH2-terminal kinase (JNK). Moreover, it can activate extrinsic pathways of apoptosis by activating the pro-apoptotic proteins, which lead to upregulation of C/EBP homologous protein (CHOP), glucose-regulated protein 78 (GRP78), inositol requiring enzyme-1(IRE1a), and JNK as observed in murine and human breast cancer cells [107].

Piperine increases phosphatidylethanolamine conjugate 3II (LC3II/ATG8), a signaling pathway that interacts with mammalian target of rapamycin (mTOR) complexes 1 and 2. These complexes are involved in cell survival, leading to their inhibition and, in turn, promoting autophagy, as seen in colon carcinoma [107].

Ferroptosis is a type of cell death that happens because of the excess accumulation of iron and lipid peroxides due to an imbalance in reactive oxygen species (ROS). Piperine has the potential to increase ROS as well as intracellular calcium levels, which can activate the Fenton reaction and lead to cell death. On the other hand, piperine activates anoikis, which is a programmed cell death that happens due to the loss of attachment to the extracellular matrix [109].

The cell cycle is a series of events that take place inside the cell in order to divide into 2 separated identical daughter cells. In cancer cells, the cell cycle is highly disrupted and irregular. The key regulators of the cell cycle include cyclin-dependent kinases (CDKs), cyclins, and CDK inhibitors (CKIs). Piperine interferes with the cell cycle by affecting various protein regulators and checkpoints, including downregulating cyclin D1 and the induction of p21, a CDK inhibitor [106].

Cancer cells usually have high levels of ROS. Multiple studies reported piperine’s ability to target ROS in two different ways depending on the administered dose. Piperine can function as a defense mechanism against ROS when given in a low dose in mouse and rat models. On the other hand, administering piperine in high doses will function as a pro-oxidant by elevating the levels of ROS, resulting in cell apoptosis [107]. Figure 8 shows the different anticancer mechanisms that piperine utilizes.

Cancer cells have a unique phenomenon that helps stabilize them and improve their chances of growth and survival; they are capable of forming new blood vessels by exciting vessels, which is known as angiogenesis or neoangiogenesis. Piperine can inhibit this crucial mechanism by inhibiting proliferation and the G1/S transition of human umbilical vein endothelial cells (HUVECs), inhibiting collagen-induced blood vessel outgrowth, downregulation of the pro-angiogenetic protein kinase B (Akt) signaling cascade, and decreasing the expression of vascular endothelial growth factor (VEGF) [110]. In addition, piperine has demonstrated the ability to induce the differentiation of osteoblasts by affecting the expression of osteogenic marker genes through AMPK phosphorylation and inhibiting the Wnt/beta-catenin signaling pathway in breast cancer [108].

Piperine is a bioavailability enhancer for several chemotherapeutic agents, such as resveratrol and curcumin. By affecting multiple pathways, piperine has an inhibitory effect on the enzymes responsible for metabolizing drugs like cytochrome P450 3A4, which alone metabolizes more than fifty percent of the marketed drugs.

Piperine-based treatments may become a standard part of the management of cancer, but further studies are needed to understand the specific molecular targets and pathways influenced by piperine, as well as an assessment of its efficacy and safety [109].

## 6. Emodin

Emodin, a naturally occurring anthraquinone derivative that is a tyrosine kinase inhibitor and is present in the rhizomes and roots of many different plants [111], is widely used in traditional Chinese medicine as a laxative (Figure 7).

Emodin has demonstrated potential in pre-clinical settings as a natural anticancer medication for the management of multiple cancer types, including breast, pancreatic, colon, stomach, liver, gallbladder, and lung malignancies [111,112,113,114]. Research has indicated that emodin suppresses the growth of various cancer cell types and inhibits pathways that lead to inflammation, proliferation, angiogenesis, tumorigenesis, invasion, and metastasis, as well as promotes mitochondrial-mediated apoptosis.

The anticancer effects of emodin were examined in A2780 and SK-OV-3 epithelial ovarian cancer cells [115]. Emodin was administered to both cancer cell lines, and the incubation period was one to three days. The antiproliferative effects of emodin therapy were dose- and time-dependent. Trans-well migration and invasion experiments further demonstrated the anti-invasive and anti-metastatic properties. This investigation also demonstrated the compound’s effects on the epithelial-mesenchymal transition. Protein analysis revealed that mesenchymal indicators like vimentin and N-cadherin were downregulated dose-dependent. In contrast, epithelial markers like E-cadherin and claudin were elevated [115]. Additionally, it was proven that the expression levels of Slug (a transcription factor) were also decreased with emodin therapy. Cells were transfected with siRNA to silence the essential proteins for Slug and β1integrin-linked kinase (ILK) to elucidate the mechanism of action. The findings showed that emodin inhibited cancer cells’ epithelial-mesenchymal transition by blocking the ILK/GSK-3β/Slug signaling [115].

SiHa, CaSki, and HaCaT cell lines showed a marked reduction in cell viability following treatment with emodin in conjunction with photodynamic therapy. Increased reactive oxygen species (ROS) generation, caspase-3 activity, and autophagic vacuole fluorescence intensity were seen in conjunction with these decreases. This indicates that apoptosis and autophagy caused cell death in cervical cancer cells due to elevated ROS generation [116]. MCF-7 human breast cancer cells were used to study emodin’s anticancer properties. The findings demonstrated that emodin could impede cell division and elevate the apoptosis rate depending on both time and dosage. According to a molecular docking study, emodin can be fixed into the ATP-binding pocket of the aryl hydrocarbon receptor (AhR) protein. Further investigation revealed that emodin can function as an AhR agonist, raising the levels of both the protein and CYP1A1, its downstream target gene [117].

These outcomes were validated with an AhR inhibitor (CH223191), showing a rise in the cell survival rate. Wang et al. conducted a study wherein they found that emodin had anticancer effects on various kidney cancer cells, including 786-0, ACHN, CAKI, and OS-RC-2. However, they had no harmful effects on HK-2 noncancerous cells. The mechanism of action was mediated by necroptosis rather than the stimulation of apoptosis. Therefore, in a manner linked to increased ROS and the consequent activation of the JNK pathway, necroptosis-related proteins such as receptor-interacting protein kinase-1 (RIP1) and mixed lineage kinase domain-like pseudo kinase (MLKL) were dramatically raised following emodin therapy [118].

Breast cancer cells’ and macrophages’ TGF-β1 production was inhibited by emodin, which also lessened the EMT and CSC development of breast cancer cells generated by TGF-β1 or macrophages. By lowering tumor-promoting macrophages and inhibiting EMT and CSC development in the primary tumors, short-term emodin treatment prior to surgery prevented breast cancer post-surgery metastatic recurrence in the lungs. Emodin blocked TGF-β1 signaling pathways in breast cancer cells, both canonical and noncanonical, and decreased transcription factors essential for EMT and CSC, according to mechanistic investigations [119].

Studies using human hepatocellular carcinoma HepaRG cells demonstrated the anticancer properties of emodin. Compared with untreated cells, emodin-induced cell cycle arrest at the S and G2/M phases and decreased cell viability in a dose/time-dependent manner. Emodin caused apoptosis in these cells and expanded the numbers of cells in both the early and late phases of apoptosis, as demonstrated by the annexin V/PI staining study [120]. The mechanism of action was associated with an increase in pro-apoptotic Bax expression and a decrease in anti-apoptotic Bcl-2 expression. Furthermore, in contrast to the vehicle-treated group, emodin administration dramatically increased the cleaved caspase-3, -9, and PARP expression levels. These results suggested that emodin may use the mitochondrial caspase-dependent route to cause apoptosis in hepatocellular cancer cells [120]. Another study also examined how emodin affected Caco-2 human colon cancer cells over the course of 24 h. Emodin suppressed cell growth (IC50) by 50% at 30 μM, resulting in a dose-dependent decrease in cell viability. Furthermore, treatment with emodin resulted in cell cycle arrest at the G2/M phase and a markedly increased proportion of cells in the early and late apoptotic stages. Furthermore, emodin was found to induce apoptosis through the mitochondrial route based on its impact on the levels of Bax/Bcl-2 protein expression and mitochondrial membrane potential. Additionally, emodin decreased the phosphorylated forms of essential proteins involved in the PI3/Akt signaling pathway, which is linked to the development of cancer and tumorigenesis. These findings raise the possibility of using emodin as an anticancer medication [121]. Emodin inhibits VEGFR2/PI3K/Akt signaling, which stops the human colon cancer cell line HCT116 from migrating, adhering, and multiplying [122].

Researchers examined emodin’s anticancer efficacy using animal models and the hepatocellular carcinoma cell line SMMC-7721. In the control group, emodin administration decreased SMMC-7721 cell proliferation in a dose and duration-dependent manner. Using the same cell line, flow cytometric analysis showed a significant correlation between the ratio of apoptotic cells and the rise in emodin concentration. The overall expression of ERK, p38, and JNK was not affected by emodin; however, the level of p-JNK was decreased, and the phosphorylated forms of ERK and p38 were markedly increased in a time-dependent manner [123].

Furthermore, there was a drop in p-Akt levels but not in total Akt expression. Moreover, the emodin-treated SMMC-7721 cells exhibited a substantial upregulation of cleaved caspase-3 and -9 expression levels. After subcutaneously injecting 5 × 10^6^ SMMC-7721 cells into BALB/c-nu nude mice, the ameliorative impact of emodin was analyzed. Treatment with emodin exhibited a dose-dependent inhibition of tumor growth in mice, with no discernible impact on the mice’s overall body weight. Emodin’s antiproliferative action in animal models was corroborated by a considerable decrease in the proliferating cell nuclear antigen (PCNA) protein and Ki-67 levels. In summary, this study showed that emodin may have ameliorative effects when used to treat hepatocellular cancer (Figure 9) [123].

A different investigation documented Emodin’s efficacy in treating human non-small lung cancer cell lines A549 and H1299. Emodin administration led to a dose-dependent decrease in the rate of proliferation and an increase in the rate of apoptosis, which is in line with the previously reported outcomes. Furthermore, tribbles homolog 3 (TRIB3)’s function was ascertained through small interfering RNA (siRNA) knockdown. The acquired data suggested that a decrease in the rate of apoptotic cells resulted from the silence of this gene. The apoptotic effect of emodin was dramatically reduced due to the suppression of NF-κB activation resulting from both TRIB3 gene silencing and 4-phenylbutyrate (4-PBA)-induced ER stress [124]. Consequently, all these results pointed to a mechanism by which ER stress and the activation of the TRIB3/NF-κB pathway in lung cancer cells drive the apoptotic effects of emodin. During the same investigation, growth factor-reduced matrigel and A549 cancer cells were administered subcutaneously into BALB/c nu/nu nude mice. When emodin treatment was combined with 4-PBA exposure, the effect was attenuated, but the tumor growth was still significantly reduced compared to the control groups [124].

Human cells T24 and T5637 were used by Ma et al. to test the action of emodin. They found that emodin inhibits the expression of Notch1, which slows the growth of bladder cancer [125]. 

## 7. Parthenolide

Sesquiterpene lactone (SL) parthenolide (PTL) was first extracted from feverfew (*Tanacetum parthenium*) shoots and has demonstrated strong anti-inflammatory and anticancer properties. Clinical trials are currently testing it for cancer [126]. Parthenolide’s structure-activity relationship (SAR) research led to the derivatization of dimethylamino-parthenolide (DMAPT), an orally bioavailable analog, by revealing important chemical features needed for biological activities and epigenetic pathways [126]. Parthenolide is the first tiny chemical discovered to be selective against cancer stem cells (CSC). It accomplishes this by focusing on particular signaling pathways and eradicating cancer from its roots [126]. The anticancer property of PTL was first validated in 1973 [127]. Furthermore, its patent application for tumor suppression was approved in 2005 [128]. Numerous studies have confirmed PTL’s antitumor potential in vitro and in vivo in various cancer types. This is primarily due to its cytotoxicity to the majority of cancer cells and its ability to specifically target cancer stem cells (CSCs), a subpopulation that is currently thought to be responsible for tumor relapse and chemotherapy resistance [129,130,131,132,133,134,135,136]. Subsequent investigations uncovered several direct PTL targets, including p65, IκB kinase (IKK), focal adhesion kinase 1 (FAK1), and others, which have an indirect impact on signaling pathways that are responsible for the antitumor properties of PTL. These pathways account for redox imbalance, cell cycle arrest, induction of apoptosis, suppression of metastasis, and epigenetic regulation [131,136,137,138].

PTL has been used in combination with several anticancer drugs, including inducers of reactive oxygen species (ROS), tubulin-directed medicines, anthracyclines, antimetabolites, histone deacetylase inhibitors, and mTOR inhibitors, as reviewed by Malgorzata et al. [137].

PTL had antiproliferative activity in practically all tests, with half-maximal inhibitory concentrations (IC50) ranging from 2.5 to 25 μM for most tumor cells, highlighting its cytotoxicity to many cancer cells [139]. Several studies have shown that PTL therapy can cause intrinsic or extrinsic apoptosis in tumor cells by activating the p53 signaling route, regulating the Bcl-2 family members, blocking the activities of the NF-κB signaling pathways, and producing reactive oxygen species [140].

According to research by Berdan et al., parthenolide covalently alters FAK1’s cysteine 427, which inhibits FAK1-dependent signaling pathways and reduces the motility, survival, and proliferation of breast cancer cells [141].

Further supporting its role as a multifunctional agent, PTL was identified as a novel inhibitor of USP7 based on the following evidence: (a) PTL inhibited USP7-mediated hydrolysis of Ub-AMC/Ub-Rho110 and di-Ub; (b) PTL competed with the binding of the Ub-VME/Ub-PA probe to USP7; and (c) CETSA, SPR, and MS analyses revealed that PTL directly interacted with USP7 [142].

In another study focusing on esophageal squamous cell carcinoma (ESCC), PTL inhibited both the in vitro proliferation and migration of ESCC cells and the tumor growth in a mouse xenograft model. Notably, PTL reduced the density of micro-vessels within the xenograft tumors and hindered the proliferation, invasion, and tube formation of endothelial cells in vitro. These effects were associated with decreased AP-1, VEGF, and NF-кB expression in ESCC cells [143].

Yuan et al. showed that PTL increased mitochondrial ROS generation and encouraged TPC-1 cell death in addition to suppressing the proliferation of thyroid cancer cells TPC-1. Cell metabonomic analysis showed that PTL administration altered the TPC-1 cells’ lipid, choline, and amino acid metabolism, suggesting that PTL can impact the cells’ TCA cycle and energy metabolism. PTL can, therefore, eventually increase apoptosis while simultaneously reducing the growth of malignant cells. PTL can act against tumors through a range of downstream targets and pathways. From a metabonomic standpoint, this work offers new insights into the anticancer effects of PTL [144].

Another study on lung cells found that PTL significantly lowered the growth of A549 and H1299 lung cancer cells. Furthermore, molecular biology research confirmed the anti-proliferative effect of PTL on lung cancer cells. Interestingly, after PTL therapy, the expression of the proliferating cell nuclear antigen (PCNA) protein significantly decreased [145].

Different dosages of parthenolide were shown by Liu et al. to decrease the growth and proliferation rates of ACHN and 786-O cells. In 786-O and ACHN cells, parthenolide treatment at 0, 4, or 8 µM produced around 170, 90, 40, 190, 150, and 70 invasive cells per field, respectively. Parthenolide inhibited the expression of MMP-2 and 9. Parthenolide treatment increased E-cadherin protein levels and decreased those of N-cadherin, vimentin, and snail. Moreover, parthenolide inhibited the PI3K/Akt pathway and cancer stem cell marker production [146].

PTL did not harm normal cells, but it did reduce the viability of C918 and SP6.5 cells in a dose-dependent manner, with PTL having a greater effect on C918 cells than on SP6.5. Second, PTL caused a drop in the proportion of cells at the S phase and an increase in the proportion at the G1 phase of the cell cycle in C918 cells, but the proportion remained unchanged at G2. Furthermore, PTL reduced the expressions of Cyclin D1, B-cell lymphoma-2 (Bcl-2), and B-cell lymphoma-extra-large (Bcl-XL), as well as triggered apoptosis in C918 cells. Additionally, PTL enhanced the production of caspase-9, caspase-syl aspartate specific proteinas-3 (caspase-3), Bcl-2-associated X protein (Bax), and cyclin inhibition protein 1 (P21). However, caspase-8’s expression remained unchanged (Figure 10).

## 8. Luteolin

One of the most studied naturally occurring flavones among chemicals obtained from plants is luteolin (3′,4′,5,7-tetrahydroxyflavone), a subclass of flavonoids made up of a C6–C3–C6 carbon skeleton with two benzene rings connected by a heterocyclic ring Its chemical appearance is that of a yellow crystalline material with a poor water solubility, molecular weight of 286.24 g/mol, and molecular formula of C_15_H_10_O_6_ [147].

Interestingly, the luteolin compound is also considered heat stable, meaning it does not evaporate when cooking. Studies of the structure-activity relationship showed that luteolin’s potent antioxidant activity results from the hydroxyl groups present at the sites of C5, C7, C3′, and C4′. Its ability to combat microbes is also ascribed to carbonyl oxygen at the C4 site. Furthermore, it has been determined that the biocidal activity of luteolin derived from the double bond between C2 and C3 [147]. Plants contain copious amounts of the luteolin molecule, which is found as an aglycone molecule without a sugar moiety and as a glycoside molecule (named LUT-7-O-glucoside or LUT-7G) with a sugar moiety, the main one being glucose. The primary distinction between luteolin’s aglycone and glycoside forms is found in their molecular structures; in the glycoside form, sugar moieties are joined by one or more hydroxyl groups. The most prevalent luteolin compound found in diets containing plant-based foods and drinks, including dark chocolate, green tea, coffee, almonds, apples, oranges, pomegranates, lemons, grapes, lettuce, spinach, seaweed, oregano, parsley, and thyme. Furthermore, when the Luteolin (aglycone form) and LUT-7-O-glucoside forms were evaluated for their respective actions, the aglycone form showed more potent antioxidant, antidiabetic, and anti-inflammatory properties [147].

Luteolin’s anticancer properties have been attributed to its ability to inhibit tumor cell invasion, metastasis, and proliferation through various mechanisms, including the induction of apoptosis, suppression of kinases, regulation of the tumor cell cycle, and reduction of transcription factor activity (Figure 11).

## 9. Quercetin

Recently, flavonoids have been introduced as diet-derived components to combat tumor cells by modulating the pathways involved in angiogenesis, apoptosis, and proliferation. Flavonoids are naturally occurring polyphenolic compounds with several demonstrated benefits, such as anti-inflammatory, anti-diabetic, antimicrobial, anticancer, antioxidant, antiviral, and anti-allergic properties [148,149]. Within the flavonoid subclass of flavonols, Quercetin (QUE; 3,5,7,30,40-pentahydroxyflavone) is the primary representative. Quercetin is a dietary flavonoid that is commonly found in fruits and vegetables and is consumed in copious quantities in Western diets. One of the vegetables that is especially high in this flavanol is onions. The content of this flavonoid in 100 g of Fresh Weight (FW) of white and yellow onions ranges from 0.03 to 0.28 mg, with red onion varieties having the highest content (about 1.31 mg/100 g FW) [150,151]. QUE is primarily found in plants as glycosides. The hydrolysis of glycosidic bonds is catalyzed by intestinal β-glycosidases prior to their absorption by enterocytes. Here, they are converted into conjugates of quercetin [152]. QUE is metabolized in the liver and intestines [153]. QUE 3-O-glucuronide and QUE 3′-O-sulfate are the two main QUE metabolites that are found in human plasma, while parent aglycon is known to be present in the systemic circulation [154]. The systemic effects of QUE are influenced by its low bioavailability, which may be responsible for the variations in QUE effects observed in vitro and in vivo [152,155]. Quercetin modulates the PI3K/Akt/mTOR, Wnt/β-catenin, and MAPK/ERK1/2 pathways to exert its anticancer effects on cancer cells and tumors. Moreover, because quercetin reduces β-catenin and HIF-1α stabilization, activates caspase-3, and inhibits Akt, mTOR, and ERK phosphorylation, it promotes cell viability loss, apoptosis, and autophagy in cancer. Additionally, quercetin inhibits metastasis by lowering MMP and VEGF secretion. Key enzymes involved in glycolysis and glucose uptake are inhibited by QUE’s metabolic effect on cancer through disruption of the PI3K/Akt/mTOR pathways. In addition to reducing bioenergetics and inducing intrinsic apoptosis, quercetin also targets mitochondria in cancer. The reduction of cell viability, inhibition of metastasis, and induction of apoptosis in cancer cells are all influenced by QUE’s effects on glucose metabolism and cellular energy production [156]. The effects of quercetin are dose-dependent and biphasic. QUE has chemopreventive effects at low concentrations because it acts as an antioxidant; however, at high concentrations, it acts as a pro-oxidant and may have chemotherapeutic effects [157]. It has been established that QUE can prevent the in vitro proliferation of several cancer cell lines. The primary cause of QUE’s anti-proliferation effect was cell cycle arrest in the G1 phase despite a low QUE dose with a slight cytotoxic effect. This has been primarily achieved by phosphorylating the retinoblastoma tumor suppressor protein, pRb, and downregulating cyclin B1 and cyclin-dependent kinase 1 (CDK1), two critical elements of G2/M cell cycle progression [158]. Hypophosphorylated Rb inhibits the expression of cell proliferation genes by binding to and sequestering the transcription factor E2F1. This causes cell cycle arrest at the G1 phase [159]. Additionally, quercetin activated Chk2 and caused minor DNA damage, which led to the induction of p21, a cyclin-dependent kinase) inhibitor [158]. QUE also prevented the progression of the cell cycle from G0/G1 to G2/M at high concentrations [160]. Moreover, Strong anti-mitotic activity was demonstrated by quercetin, which reduced the activity of several kinases (including platelet-derived growth factor (PDGF), MET kinase, NIMA-related kinases (NEK4 and NEK9), Aurora kinases A and B, and PAKs (p21-activated kinases) by more than 80% [161]. It is noteworthy that QUE exhibits this effect at a low concentration (2 µM), which is less than 10% of its IC50 growth-inhibitory concentration, as determined by averaging eight different cancer cell lines (the mouse melanoma cancer cell line, human non-small cell lung cancer, glioblastoma, colon cancer, breast and prostate cancer cell lines, and melanoma) [161]. Moreover, Pre-treatment with 300 µM QUE significantly down-regulated the phosphorylation of Akt, PDK1, Bcl-2-associated death promoter (BAD), and the amount of tumor necrosis factor receptor 1 (TNFR1). It also suppressed the increase in H_2_O_2_-induced ROS. Additionally, in Dalton’s lymphoma ascite (DLA) cells treated with H_2_O_2_, QUE raised the amount of PTEN [161]. Furthermore, some research indicates that QUE’s anti-tumor effects stem from its capacity to trigger autophagy and apoptosis in cancerous cells and xenographmodels. The number of cells in the sub-G1 phase, nucleus fragmentation, activation of caspase-3 and caspase-9, and poly (ADP-ribose) polymerase protein degradation are all enhanced by quercetin. Quercetin also decreased the potential of the mitochondrial membrane in malignant glioma cells U373MG. Quercetin causes intrinsic apoptosis by activating JNK, increasing p53 expression, and translocating it to mitochondria [162]. Moreover, quercetin can trigger protective autophagy in breast and gastric cancer cells by inactivating the Akt-mTOR pathway and HIF-1α signaling [163]. Quercetin has also increased G1 phase cells and caused apoptosis, promoting PARP cleavage and nuclear fragmentation.

Furthermore, QUE has suppressed the phosphorylation of Aktser473 and mTOR, encouraged the dephosphorylation and activation of glycogen synthase kinase 3 (GSK-3), decreased the expression of pro-survival cellular proteins like c-FLIP, cyclin D1, and c-Myc, and caused the degradation of β-catenin. Additionally, QUE has reduced the phosphorylation/activation of the signal transducer and activator of transcription 3 (STAT3) Tyr705/Ser727, as well as the release of interleukin-6 (IL-6) and IL-10. The activation of STAT3 and PI3K/Akt/mTOR is mediated by these two cytokines [164]. The Wnt/β-catenin pathway is connected to PI3K/Akt/mTOR signaling because Akt phosphorylates GSK-3, which in turn causes its inactivation and β-catenin accumulation [165]. These findings suggest that QUE causes cell death by inhibiting the PI3K/Akt/mTOR and STAT3 pathways in PEL cells (Figure 12) [164].

## 10. Anthocyanins

Fruits and vegetables naturally contain glycosides called anthocyanins with low biosafety and cytotoxicity levels. Anthocyanins are a unique subclass of flavonoids that give various fruits and vegetables their red, purple, and blue color. This, in turn, enhances their appearance, one of the attributes consumers value the most. Beyond this business standpoint, anthocyanins have also been suggested as therapeutic agents for the prevention of cancer, heart disease, and some metabolic conditions like type 2 diabetes and obesity [166,167]. Over 500 distinct types of anthocyanins have been identified to date, and these anthocyanins are found in 72 genera and 27 families of plants [168]. Anthocyanins work as bacteriostatic, antioxidant, anti-inflammatory, anti-aging, and anticancer agents. 2-phenylchromenylium, also known as flavylium, is the basic structure of anthocyanin. Due to the different substituent groups of the B-ring on the basic structure, it has been discovered that the combination with glucose, galactose, and rhamnose could form a variety of anthocyanins, including pelargonidin, delphinidin, petunidin, cyanidin, and malvidin [169,170]. Studies have revealed that the active site, which in turn prevents tumor growth and metastasis, is the ortho-dihydroxyphenyl structure on the B-ring [171,172]. Moreover, Anthocyanins have been reported to have a variety of potential anti-tumor effects, which are related to antioxidant, anti-inflammation, anti-mutagenesis, induction of differentiation, inhibition of proliferation through modulation of signal transduction pathways, induction of cell cycle arrest and stimulation of apoptosis or autophagy of cancer cells, anti-invasion and anti-metastasis, reversal of drug resistance in cancer cells, and enhancement of chemotherapy sensitivity [173]. Moreover, Anthocyanins have a potent antioxidant potential because they can scavenge free radicals, which work to lower DNA damage and prevent the growth of tumors [174]. In addition, According to reports, anthocyanins have the ability to regulate the expression and secretion of inflammatory factors by blocking the transcription factor nuclear factor kappa-light-chain-enhancer of activated B cells (NF-κB) via a variety of mechanisms [175,176]. For instance, by acting on the PI3K/PKB and MAPK pathways, cyanidin-3-glucoside, delphinidin-3-glucoside, and petunidin-3-glucoside block the activation of NF-κB induced by external stimuli (such as lipopolysaccharide (LPS) or interferon-γ (IFN-γ)) [177,178], and can prevent the synthesis of prostaglandin E (PGE2) and nitric oxide (NO), as well as the expression of cyclooxygenase 2 (COX-2) and inducible nitric oxide synthase (iNOS) [176]. However, depending on the various substituents on their B rings, anthocyanins have varying anticancer effects. A growing body of research suggests that the primary molecular mechanism underlying their anti-tumor effects is their inhibition of cancer cell growth and metastasis through targeting RTKs (EGFR, PDGFR, and VEGF/VEGFR). Moreover, interacting with the Ras-MAPK and PI3K/Akt signal cascade pathways/pathways. At the initial stage, anthocyanin inhibits inflammation by suppressing the expression of COX-2 and iNOS through the PI3K/Akt and NF-κB pathway. This would prevent the normal cells from transforming by controlling the expression of phase II antioxidant enzymes to achieve anti-oxidation through the Nrf2/ARE signal system eventually. During the formation phase, anthocyanins would prevent carcinogenesis by targeting the MAPK and AP-1 pathways and inhibiting RTK activity and its signal cascade pathway, which in turn regulates the expression of cancer-related genes and causes cell cycle arrest and DNA repair. During the developmental stage, anthocyanins trigger caspase activation in cancer cells, facilitated by ROS and JNK/p38-MAPK (Figure 13). Moreover, anthocyanins prevent the metastatic growth of cancer by interfering with the VEGF signaling pathway and the breakdown of the extracellular matrix (ECM). Moreover, anthocyanins can increase cancer cells’ sensitivity to chemotherapy by reversing their multidrug resistance [172]. 

## 11. Conclusions

In conclusion, studying natural products in the context of altered metabolism in cancer represents a captivating and promising frontier in cancer research. The intricate interplay between cellular metabolism and cancer progression has led researchers to explore the therapeutic potential of various natural compounds derived from plants, fruits, and other sources. These compounds exhibit a wide range of bioactive properties, influencing diverse metabolic pathways crucial for cancer cells’ survival and proliferation.

One of the remarkable aspects of natural products is their ability to modulate energy metabolism in cancer cells. Compounds such as resveratrol, found in grapes, and curcumin, derived from turmeric, have demonstrated the capacity to interfere with glycolysis, the Warburg effect, and mitochondrial function. By disrupting these fundamental processes, natural products have shown the potential to impair the energy production mechanisms that fuel the uncontrolled growth of cancer cells.

Moreover, the impact of natural products extends beyond energy metabolism. Compounds like quercetin and sulforaphane have been found to influence cellular redox balance and oxidative stress, contributing to the selective targeting of cancer cells while sparing normal cells. These antioxidants counteract the heightened oxidative stress characteristic of cancer cells, presenting a novel avenue for therapeutic intervention (Table 1).

Natural products also exhibit the ability to interfere with biosynthetic pathways critical for cancer cell survival. Polyphenols, such as epigallocatechin gallate (EGCG) from green tea, have been shown to modulate lipid metabolism and inhibit enzymes involved in nucleotide synthesis. By disrupting these biosynthetic processes, natural products exert anti-proliferative effects, hindering the uncontrolled cell division typical of cancer.

The complexity and heterogeneity of cancer metabolism demand multifaceted approaches, and natural products offer a rich source of diverse bioactive compounds. The synergistic effects of various constituents within these natural products present a holistic strategy that can target multiple aspects of altered metabolism in cancer cells. While challenges and questions remain, ongoing research endeavors aim to unravel the precise mechanisms of action, optimize dosages, and explore potential synergies between natural compounds and conventional cancer therapies.

Exploring natural products in cancer metabolism research underscores the importance of integrating traditional knowledge with modern scientific approaches and opens avenues for innovative and personalized therapeutic strategies. As we uncover the intricacies of altered metabolism in cancer, natural products stand poised as valuable allies in pursuing effective and holistic cancer treatments.

## Figures and Tables

**Figure 1 ijms-25-09593-f001:**
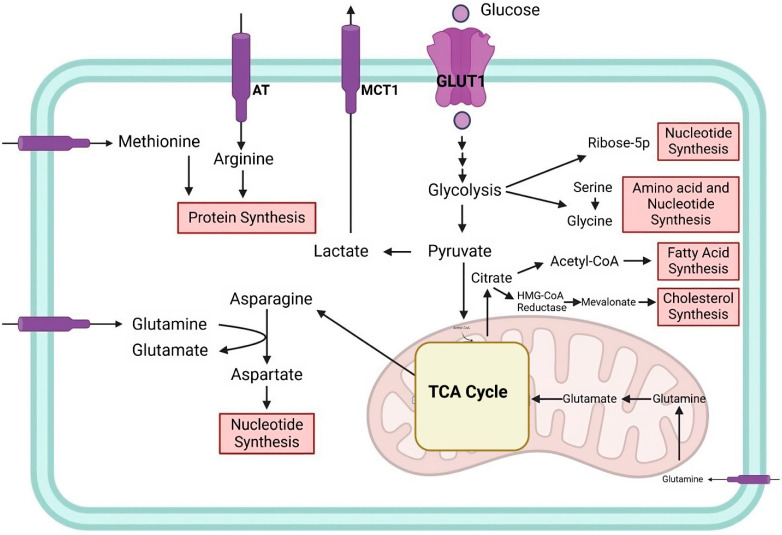
The Main metabolic pathways deregulated in cancers. This figure illustrates the reprogrammed metabolic pathways found in cancer cells. It emphasizes the Warburg effect, which results in lactate production even in oxygen since glucose is primarily metabolized through glycolysis. Important intermediates that support the production of fatty acids, cholesterol, proteins, nucleotides, and acetyl-CoA include pyruvate. The other amino acids, such as aspartate and asparagine, aid in cellular growth, whereas glutamine metabolism powers the TCA cycle and nucleotide production. The diagram shows how these changes in metabolism are coordinated by cancer cells to maintain their fast growth and survival.

**Figure 2 ijms-25-09593-f002:**
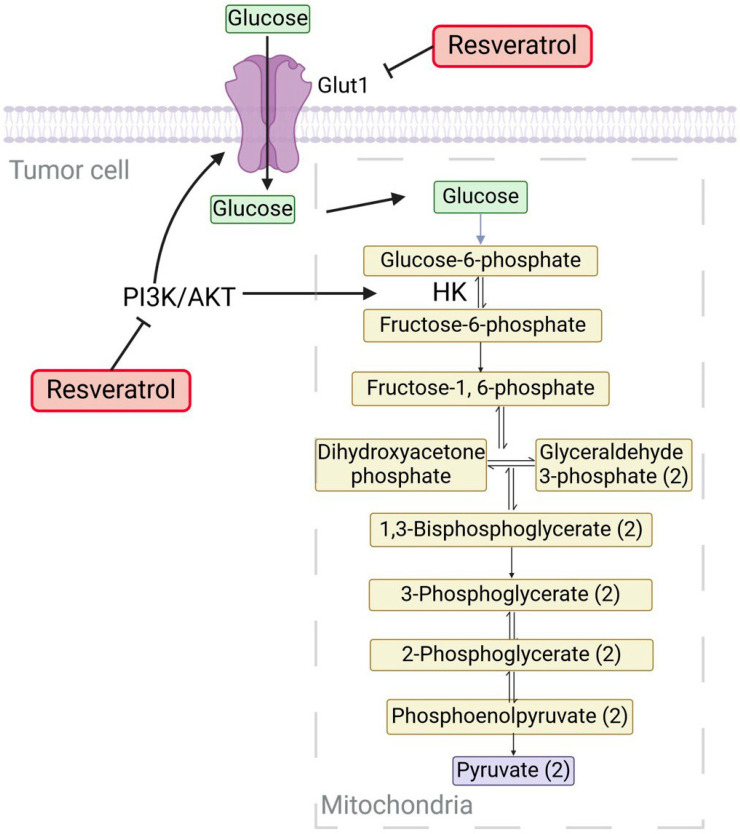
Resveratrol appears to inhibit necessary glycolysis-related enzymes, causing a change in cellular energy metabolism to oxidative phosphorylation and encouraging a more normal metabolic state. This metabolic reprogramming may prevent cancer cells from proliferating quickly by limiting their access to energy.

**Figure 3 ijms-25-09593-f003:**
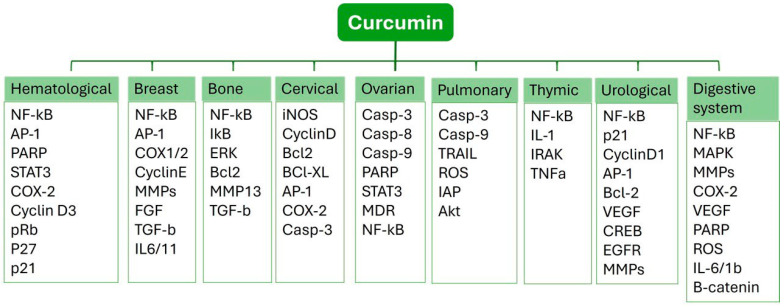
Illustration of multiple molecular targets of curcumin in various cancers.

**Figure 4 ijms-25-09593-f004:**
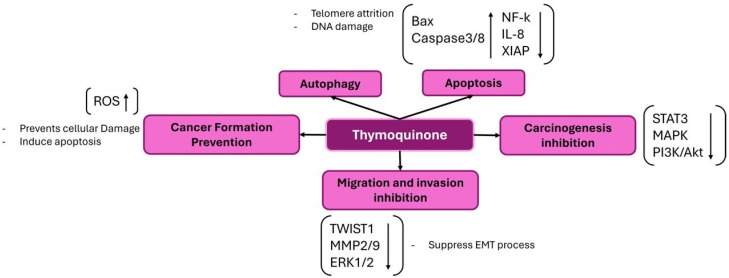
The figure demonstrates the several ways that thymoquinone works to prevent cancer. Thymoquinone inhibits cancer development by raising reactive oxygen species (ROS) levels, which trigger apoptosis and prevent cellular damage. Furthermore, it inhibits cancer development by modifying vital signaling pathways like STAT3, MAPK, and PI3K/Akt. Furthermore, it triggers apoptosis by controlling proteins linked to apoptosis, such as Bcl-2, Bax, and Caspase-3/8. Thymoquinone also suppresses the migration and invasion of tumor cells by downregulating EMT-related factors such as MMP2/9, ERK1/2, and TWIST1.

**Figure 5 ijms-25-09593-f005:**
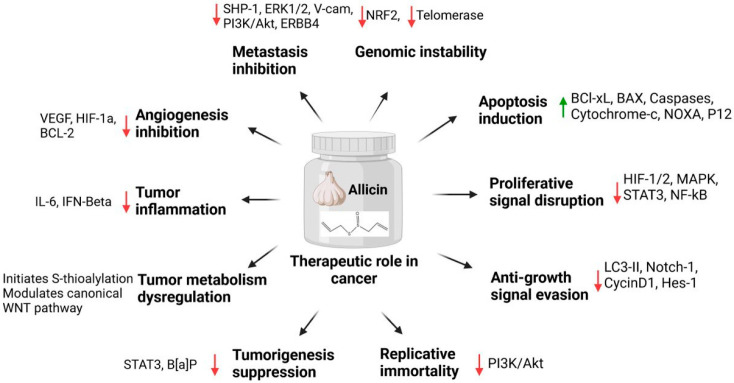
The figure shows that allicin can stimulate apoptosis and interfere with proliferative signaling pathways such as HIF-1/2, MAPK, and NF-κB while suppressing metastasis, genomic instability, and angiogenesis. Additionally, pathways, including STAT3 and PI3K/Akt, suppress carcinogenesis, modify tumor metabolism, and lower tumor inflammation. Allicin also inhibits replicative immortality and avoids anti-growth signals, which adds to its overall anticancer effect.

**Figure 6 ijms-25-09593-f006:**
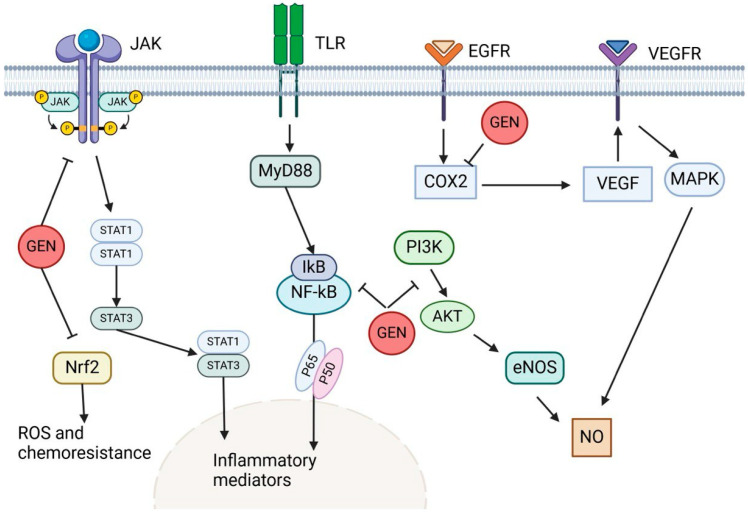
This figure illustrates how genistein (GEN) targets multiple signaling pathways in cancer metabolism and therapy resistance. By blocking the JAK/STAT pathway, GEN inhibits the generation of ROS and chemoresistance. Additionally, it suppresses the TLR/MyD88/NF-κB pathway, reducing inflammatory mediators connected to tumor development. Furthermore, GEN suppresses COX-2, which impacts the PI3K/Akt pathway and lowers angiogenesis and cell survival. GEN, the last target, inhibits VEGF-driven angiogenesis by targeting the VEGFR/MAPK pathway.

**Figure 7 ijms-25-09593-f007:**
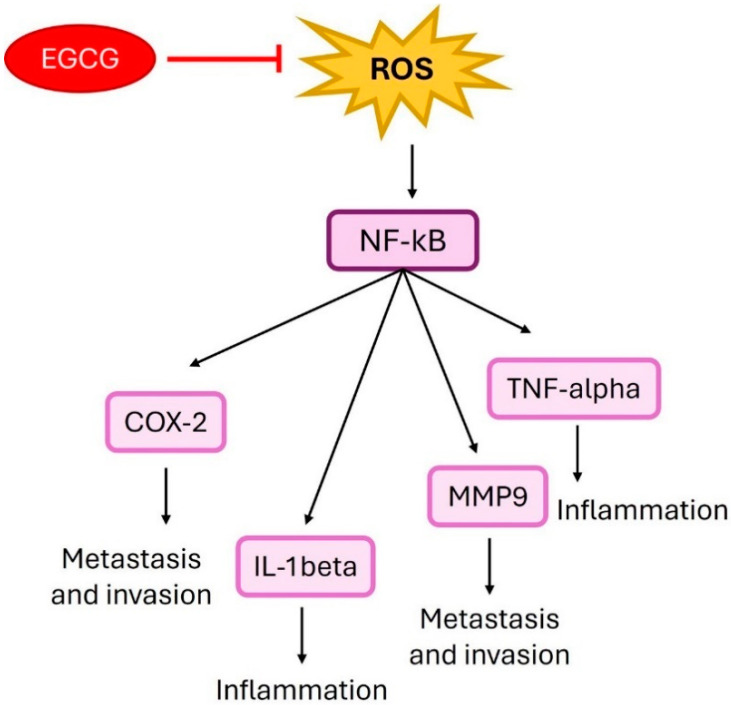
The figure depicts how epigallocatechin gallate (EGCG) suppresses reactive oxygen species (ROS) and its downstream effects on NF-κB signaling. EGCG inhibits ROS, which in turn reduces NF-κB activation. This inhibition decreases COX-2, IL-1β, and MMP9 expression, reducing metastasis, invasion, and inflammation. Additionally, the suppression of TNF-α by NF-κB further diminishes inflammatory responses.

**Figure 8 ijms-25-09593-f008:**
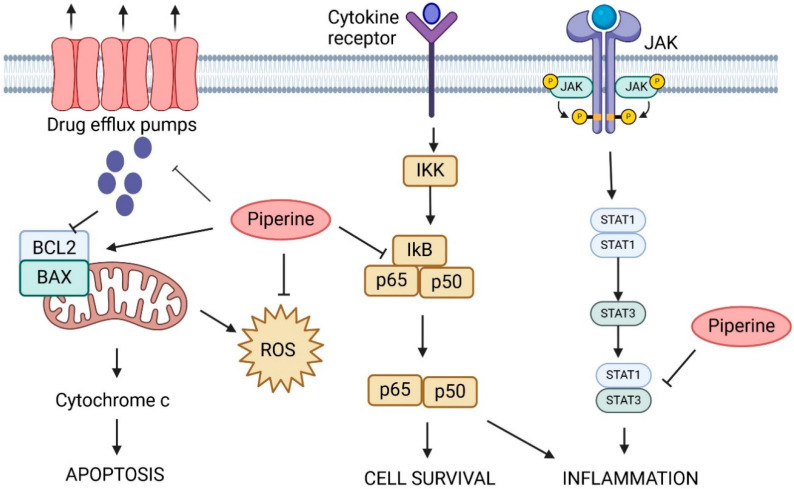
In this figure, piperine inhibits ROS, leading to reduced NF-κB activation, which diminishes cell survival and inflammation. It also disrupts STAT3 signaling, further reducing inflammation. By affecting the BCL2/BAX balance, Piperine promotes cytochrome c release and apoptosis. Additionally, piperine’s influence on drug efflux pumps suggests its potential to enhance chemotherapeutic efficacy.

**Figure 9 ijms-25-09593-f009:**
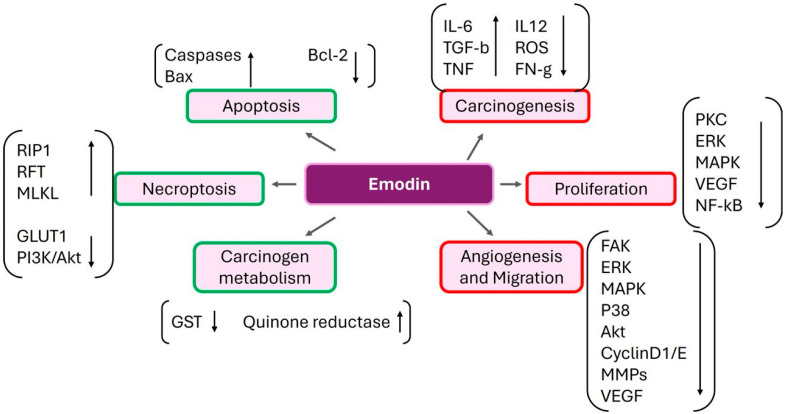
The figure illustrates how cancer processes are modulated by emodin. It promotes apoptosis and necroptosis while inhibiting proliferation, angiogenesis, and migration. It also enhances carcinogen metabolism and suppresses carcinogenesis by regulating critical pathways like FAK, ERK, MAPK, VEGF, and inflammatory cytokines.

**Figure 10 ijms-25-09593-f010:**
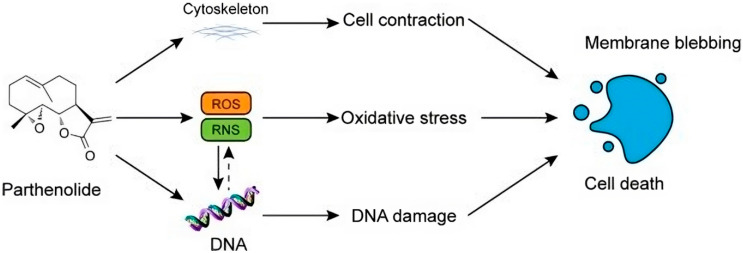
The figure shows how parthenolide induces oxidative stress through the generation of ROS and RNS, leading to DNA damage, cell contraction, and membrane blebbing, ultimately resulting in cell death.

**Figure 11 ijms-25-09593-f011:**
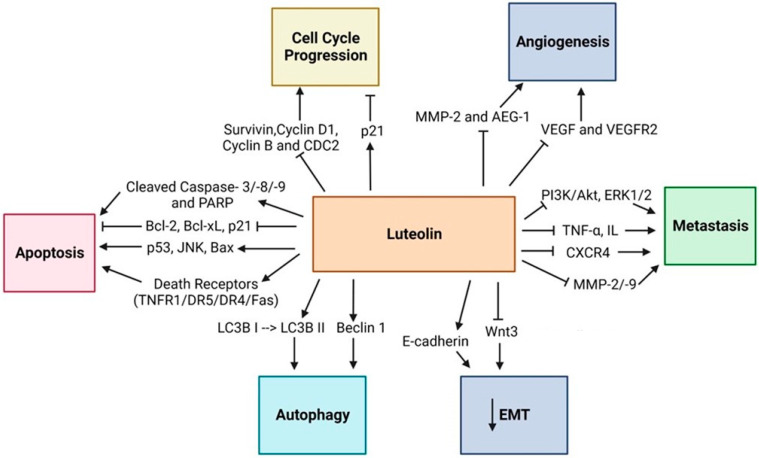
The figure illustrates luteolin’s role in inducing apoptosis by activating caspases and downregulating Bcl-2 while promoting autophagy via LC3B and Beclin 1. Luteolin inhibits cell cycle progression, angiogenesis, and metastasis by targeting key molecules such as Cyclin D1, MMP-2, VEGF, and the PI3K/Akt pathway. Additionally, it suppresses epithelial-mesenchymal transition (EMT) by modulating E-cadherin and N-cadherin, highlighting its therapeutic potential in cancer.

**Figure 12 ijms-25-09593-f012:**
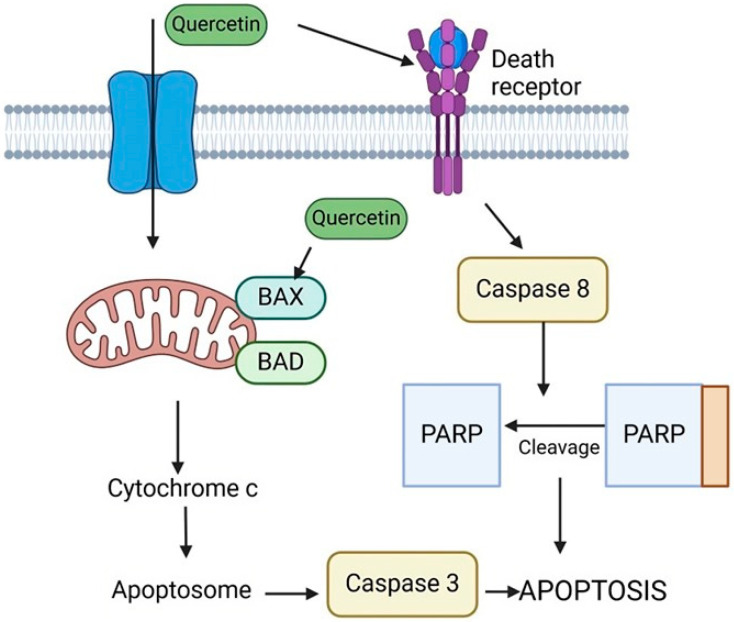
This figure illustrates the mechanism by which quercetin induces apoptosis in cancer cells. Quercetin activates death receptors, leading to the activation of caspase-8. This activation triggers the cleavage of PARP and the subsequent activation of caspase-3. Concurrently, Quercetin promotes the release of cytochrome c from the mitochondria by upregulating pro-apoptotic proteins BAX and BAD, further driving apoptosome formation and caspase-3 activation, ultimately resulting in apoptosis.

**Figure 13 ijms-25-09593-f013:**
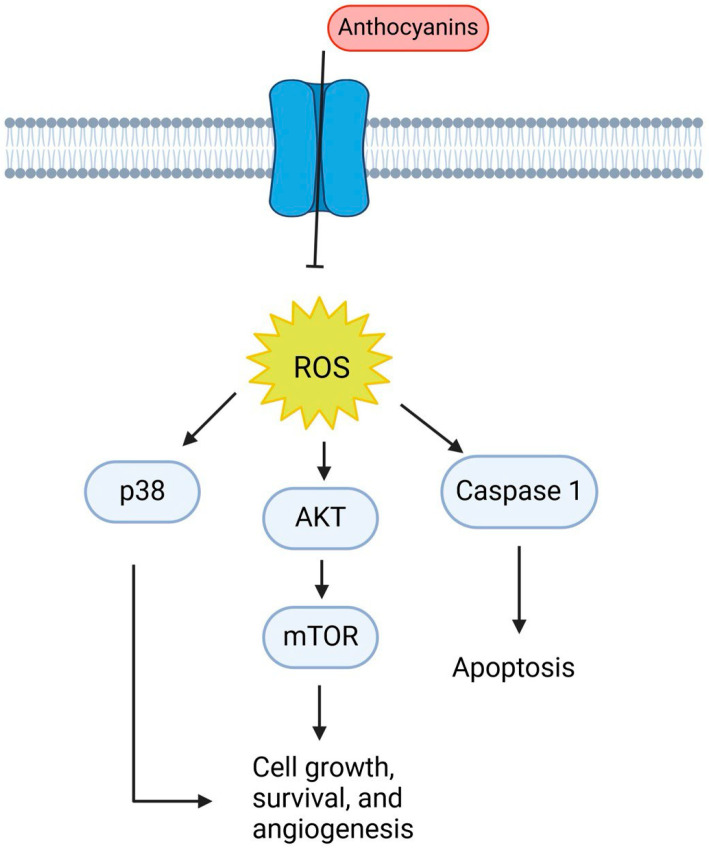
The figure shows how anthocyanins induce the production of reactive oxygen species (ROS), activating multiple signaling pathways. ROS activation leads to the phosphorylation of p38 and Akt, with downstream effects on mTOR, promoting cell growth, survival, and angiogenesis. Concurrently, ROS triggers caspase-1 activation, leading to apoptosis.

**Table 1 ijms-25-09593-t001:** Summary of the primary effects of natural products in cancer, with a focus on cancer metabolism.

Natural Product	Compound Structure	Concentration Used	Model	Mechanism of Action	Reference
Resveratrol	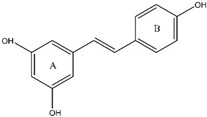	1, 5, 15, 50 or 100 μM	MCF-7 (breast cancer)	Resveratrol suppresses HIF-1α buildup, impairing glucose metabolism and lowering breast cancer cell survival. It also directly inhibits PFK activity, GLUT1-mediated glucose absorption, and intracellular ROS.	[179]
		5–150 µM (IC50 ~ 80 µM)	SKBR-3 (breast cancer)	In SKBR-3 cells, Resveratrol down-regulates FASN and HER2 genes, triggering apoptosis. Resveratrol inhibits the Akt/PI3K/mTOR pathway by suppressing Akt phosphorylation. Resveratrol inhibits FASN function, which down-regulates HER2 production.	[65]
		0, 20 or 50 μM	BEP2D cells (breast cancer)	Resveratrol inhibits glycolytic flow and GLUT1 expression by targeting ROS-mediated HIF-1α activation.	[180]
		(0, 25, 50, 100) μM	NSCLC cells. (lung cancer)	Resveratrol causes A549 cancer cells to undergo autophagy through the upregulation of glucosylceramidase beta1 (GBA1), the gene linked to Gaucher disease that produces glucocerebrosidase, an enzyme that breaks down glucosylceramide into ceramide and glucose.	[181]
		200 μM	A549 cells (lung cancer)	Resveratrol modulates polycyclic aromatic hydrocarbon metabolism genes, potentially preventing lung cancer by inhibiting CYP1A1 and 1B1 enzymes and overexpressing microsomal epoxide hydrolase, altering carcinogenic metabolite synthesis.	[182]
		10 µM	PCA model (colorectal cancer)	Resveratrol targets the pyruvate dehydrogenase complex in colorectal cancer cells, modulating lipidomic activity, increasing oxidative activity, and reversing the Warburg effect by decreasing glycolysis and pentose phosphate activity.	[11]
		50–150 μM	HT-29 colon cancer cells	Resveratrol induces apoptosis by targeting the talin-FAK and pentose phosphate signaling pathways.	[183]
		50 mM	LLC, HT-29, and T47D cells (Colorectal Cancer)	Resveratrol reduces glucose absorption by targeting ROS-induced HIF-1α activation.	[184]
		1, 5, 10, 20, and 50 µM	HCT116 SW480 (Colorectal Cancer)	Resveratrol increases Sirt1 protein expression in CRC cells, but Sirt1 knockdown eliminates its inhibitory effects on cell viability and proliferation. In CRC cells, Sirt1 knockdown prevents resveratrol-induced suppression of NF-κB-dependent gene products associated with proliferation and metastasis.	[185]
		50 µM	TRAMP cells (prostate cancer)	Resveratrol promotes cell death by increasing ROS formation, expressing apoptotic biomarkers (Bax, p53, HIF-1α), and inhibiting the anti-apoptotic protein Bcl2. It induces apoptosis in prostate cancer cells through the HIF-1α/ROS/p53 signaling pathway.	[186]
		(0–25) µM	LNCAP cells (prostate cancer)	Resveratrol inhibits HIF-1α, reducing nuclear β-catenin protein and suppressing the transcriptional activity of androgen receptor (AR) signaling, thereby inhibiting tumor development and inducing apoptosis in CRPC.	[187]
		50 and 100 μM	A2780 cells (ovarian cancer)	Resveratrol significantly reduces glucose absorption and lactate generation while lowering phosphorylated Akt and mTOR, which promote glucose uptake and glycolysis.	[42]
		0–50 μM	WSU-CLL and ESKOL cells (B-CLL)	Resveratrol induces apoptosis in B-cell lines and B-CLL patients by affecting annexin V binding, caspase activation, DNA fragmentation, and mitochondrial transmembrane potential. It also inhibits iNOS protein expression and NO release, showing potential for B-CLL therapy.	[188]
		80–200 µM	HepG2, Bel7402 and SMMC7721 (Hepatocellular Carcinoma)	Resveratrol induces apoptosis by inhibiting the PI3K/Akt pathway and lowering FoxO3a phosphorylation. Resveratrol activates SIRT1 and prevents post-translational changes in PI3K/Akt signaling. Resveratrol reduces cell migration and increases DLC1 phosphorylation by Akt.	[68]
		50 μM	Panc-1 cells (Pancreatic Cancer)	Resveratrol prevents Panc-1 cell proliferation induced by hyperglycemia. Resveratrol reduces hyperglycemia-induced ROS and H_2_O_2_ generation. Resveratrol inhibits the ERK and p38 MAPK pathways activated by hyperglycemia. Resveratrol prevents pancreatic cancer cell migration and invasion when exposed to H_2_O_2_.	[189]
		Not mentioned	MIA Paca-2&PANC 1 (Pancreatic Cancer)	Resveratrol suppresses HIF-1α and prevents HIF-1α-mediated cellular abnormalities in Pancreatic Cancer cells, as demonstrated by MDS and in vitro studies.	[190]
		50 and 100 µM	MIA PaCa-2 (Pancreatic Cancer)	Resveratrol increases sensitivity to Gemcitabine by blocking lipid synthesis through SREBP1, reducing sphere formation, stemness, and CSC marker expression.	[191]
		6.25, 12.5, 25, 50, 100, 200, and 400 μM	MGC803 (gastric Cancer)	In MGC803 cells, resveratrol suppresses the cyclin D1 signaling pathway that is dependent on GSK3β. In MGC803 cells, resveratrol suppresses PI3K/Akt signaling. Resveratrol inhibits the PI3K/Akt pathway by controlling PTEN activity.	[68]
		0–300 µM	GSCs derived from human biopsies. (Glioblastoma Cancer)	change in cell shape following RES activation of GSC necrosis at dosages greater than 150 µM of resveratrol. Resveratrol had no effect on the cell shape or proliferation of human NSCs nor on their viability. SIRT2 activity blockage or downregulation of SIRT2 expression using siRNAs mitigated the proliferative effect of RES.	[192]
Curcumin		0–100 μM	Wild-type and Bcl-2 + MCF-7 (breast cancer)	It has been proposed that curcumin inhibits the PI3K/Akt signaling pathway in breast cancer cells to cause apoptosis and autophagy.	[193]
		5, 10, 20 and 40 µM	A549 cells (lung cancer)	Through the overexpression of microRNA-192-5p and the inhibition of the PI3K/Akt signaling pathway, curcumin decreased the growth of human non-small cell lung cancer cells and induced apoptosis.	[194]
	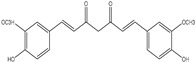	(CUR, 50 or 100 mg/kg p.o.)	positive model rats with TNBS-induced colitis (Colorectal cancer)	It was demonstrated that curcumin’s apoptotic effects were mediated by the production of ROS and the reduction of the transcriptional activities of JNK, P38, MAPK, and AP-1. Moreover, curcumin was found to reduce PGE2 expression, which eventually caused apoptosis and stopped the cell cycle.	[195]
	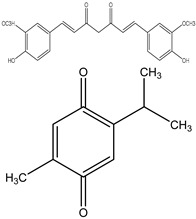	0–50 μM	Akt (Burkitt’s lymphoma)	Inhibition of the PI3K/Akt-dependent NF-κB pathway.	[85]
		10–40 μM	HEP3B, SK-Hep-1 and SNU449 cells. (Hepatocellular Carcinoma)	Curcumin interferes with Notch-1 signaling in the HEP3B, SK-Hep-1, and SNU449 cell lines within the Notch intracellular domain. Studies on the pharmacological effects of curcumin in carcinogenic-induced HCC have been carried out both in vitro and in vivo. Through decreased expression of p21-Ras, P53, and NF-κB, curcumin protected animals against diethylnitrosamine (DENA)-induced hyperplasia and HCC.	[196]
		5 μM	AGS cells (gastric cancer)	It has been shown that curcumin acts by inhibiting the Akt/mTOR/p70S6 signal pathway and activating caspase-3, a mediator of apoptosis.	[197]
		20 μM	NTera-2 cells/human (Testicular germ cancer)	Following curcumin incubation, there was a decrease in Cis 18:2, MUFA, and PUFA and an increase in trans 16:1 and 18:2, SFA, SFA/MUFA. Trans 16:1, 20:4, and trans 18:2 were higher and lower, respectively, after bleomycin and curcumin were incubated together as opposed to bleomycin alone.	[198]
		50 μM	oesophageal cell lines (OE33)	curcumin enhanced apoptosis and inhibited bile-induced NF-κB activity and DNA damage in vivo, indicating that it may be used as a chemopreventative in the rat esophagus.	[199]
		10, 15, 30 μM	C6 glioma (Brain cancer)	Curcumin increased mitochondrial dysfunction accompanied by caspase-3 activation, downregulated Bcl-xL, and reduced PI3K/Akt and NF-κB activation.	[200]
		2, 4, 6, and 8 μM	CPT-11-R LoVo colon cancer cells	Stress-activated protein kinases JNK and p38 were found to be activated by TQ; it was thought that these kinases caused TQ-induced ACD. The 8 μM TQ cleaved caspase-3 and decreased cytochrome c.	[87]
Thymoquinone	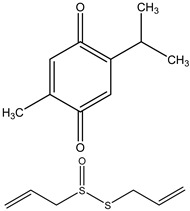	23, 36, 77 µM respectively	MIA PaCa-2, PANC-1, and FR2 (Pancreatic Cancer)	There was a decrease in expression of PKM2 when treated in combination doses. There was a decrease in Pro-Caspase-3 followed by a decrease in PARP. A decrease in the pyruvate kinase activity in treated cells followed by a greater decrease in combination.	[88]
		(0, 40, 60, and 80) μM	bladder cancer cell lines (T24 and 253J) (Bladder cancer)	TQ activates caspase to cause human 1 cancer cells to undergo apoptosis. In bladder cancer, TQ causes mitochondrial malfunction and triggers the mitochondrion-mediated apoptotic pathway. TQ, in a dose-dependent way, enhances downstream molecules, including p-eIF2a and ATF4, as well as the expression of PERK, IRE1, and ATF6. As a feedback effect, GRP78 (BIP), which is linked to PERK, IRE1, and ATF6 on ER, is also deregulated.	[86]
		(0–70) μM	Neuro-2a &normal cells (neuroblastoma)	TQ causes the release of cytochrome c from the mitochondria and depolarization of the mitochondrial membrane.	[201]
		0, 5, 10 and 20 μg/mL TQ	CEMss cells (ALL)	TQ raises the level of ROS on CEMss cell lines by acting as a concentration-dependent stimulator of ROS generation. Overexposure to ROS can result in oxidative damage to proteins, lipids, and DNA, which can promote tumor development or cell death.	[89]
		Allicin 10 µg/mL + cisplatin 2 µg/mL for 24 h	A549	Allicin increases apoptosis in a ROS-dependent way by enhancing the growth-inhibitory effect of cisplatin on A549 cells. Allicin induces p21, which in turn triggers ROS-dependent and p53-mediated cell cycle arrest.	[202]
Allicin	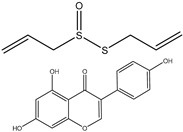	Allicin 3–10 µg/mL + 5-FU 100–300 µg/mL for 48 h	SK-Hep-1	In vitro, 5-fluorouracil synergistically sensitizes hepatocellular carcinoma cells when combined with allicin. Allicin increases the ROS-mediated mitochondrial pathway that triggers 5-FU-induced apoptosis.	[203]
		Allicin 3–10 µg/mL + 5-FU 100–300 µg/mL for 48 h	BEL-7402	Same result as above	[203]
		Allicin 10 µg/mL + dexamethasone 50 µM 24–72 h	Side population sorted from RPMI-8226 and NCI-H929 cells	In SP cells, cotreatment with DATS+Dex suppresses PI3K/Akt/mTOR activation. The effects of cotreatment with diallyl thiosulfinate and dexamethasone (DATS + Dex) on the phosphoinositide 3-kinase (PI3K) pathway of multiple myeloma (MM) side population (SP) cells are reversed by silencing miR-127-3p expression.	[204]
		Endothelial cells of mouse aorta	2.5, 5, 10, 20, and 50 µg/mL	Cell viability is increased, superoxide dismutase (ROS) levels are decreased, protein levels of 8-OHDG are decreased, NF-κB is decreased, Nox4 is decreased, HIF-1α is decreased, mRNA expression of NF-κB is decreased, Nox4 is decreased, and HIF-1α is decreased by PKC pathway and the regulation of HIF-1α.	[205]
		Human umbilical vein endothelial cells (HUVECs)	2.5, 5, 10, and 20 ng/mL; 5 ng/mL	Cell viability is increased, phosphorylation and activity of Sirt1 are increased, protein expression of plasminogen activator inhibitor-1 (PAI-1) is decreased, and the aging process of HUVECs is decreased by its activity on Sirt-1.	[206]
		Nucleus pulposus (NP) cells	0, 5, 10, 20, and 40 μM; 0, 5, 10, and 20 μM	Oxidative stress and mitochondrial dysfunction are decreased by its effect on the P38-MAPK signaling pathway.	[207]
		Mouse osteoblast MC3T3-E1 cells	0, 10, 30, and 100 µg/mL	ROS production and apoptosis are decreased, and mitochondrial dysfunction is decreased by its effect on PI3K/Akt and CREB/ERK signaling pathways.	[208]
		Adult BALB/c mice	1, 5, and 10 mg/kg	BBB (Basso, Beattie, and Bresnahan) scores are increased, spinal cord water content is decreased, heat shock protein 70 (HSP70) protein levels are increased, Akt phosphorylation is increased, expression of iNOS protein is decreased, ROS levels are decreased, NADH levels are increased (HSP70, Akt, and iNOS signaling pathway).	[209]
		HepG2 cells	50, 100, and 200 μM	Palmitic acid (PA)-induced hepatocyte injury is decreased, and lipid accumulation is decreased by the PPAR signaling pathway.	[210]
		High-fat-diet (HFD)-induced obesity and genetically leptin-receptor-deficient obese (Db/Db) mouse models	4.2, 2.52, and 1.26 mg/day/mouse; 0.1, 1, 10, 50, and 100 μM; 50 μM	Body weight gain is decreased, adiposity is decreased, glucose homeostasis is maintained, insulin resistance is increased, and hepatic steatosis is decreased. Brown adipose tissue (BAT) activation, Sirt1-PGC1α-Tfam pathway, succinylation levels of UCP1 in BAT are increased, sirt5.	[211]
		Adult male Wistar rats	20 mg/kg wt	Total lipid levels are decreased, levels of oxidized low-density lipoprotein (ox-LDL) are decreased, lipid peroxidation is decreased, and NF-κB activity is decreased.	[212]
		40 μM of genistein for 0, 3, or 24 h, respectively.	Breast BCSCs (In vitro)	CD44+/CD24−/ESA+ are decreased, PI3K/Akt is increased, MEK/ERK is increased, G2/M cell cycle arrest is increased, apoptosis is increased, BRCA1 is increased, ATR complex is increased, DNA damage is increased, TNBC is decreased, cell proliferation is decreased.	[213]
Genistein		0, 25, 50, and 75 μM exposure for 48 h.	Lung H446 (In vitro)	Apoptosis is increased, cell proliferation is decreased, FOXM1 protein is decreased, and proliferation is decreased.	[214]
		10-mM.	Liver BNL CL2, Huh7, HepG2, HA22T (In vitro)	MMP-9 is decreased, NF-κB is decreased, P-1 is decreased, AP-1 is increased, JNK is decreased, ERK is decreased, NF-κB is decreased, and phosphatidylinositol/ERK3-kinase/Akt is increased. Metastasis is decreased.	[215]
		100 μM and 200 μM.	Prostate C4-2B, LNCaP (In vitro)	Cell proliferation is decreased, and apoptosis is increased.	[216]
	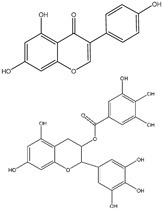	(10^−14^) to (10^−6^) M	Brain Neuroblastoma SK-N-SH (In vitro)	Cell cycle arrest at phase G2/M is increased, proliferation is decreased, Akt is increased, CHD5 is increased, and p53 is increased.	[217]
		140 mg/kg	Colon LoVo SW480, HCT116 (In vitro)	Apoptosis is increased, and NF-κB is decreased. G2/M cell cycle arrest is increased, p21waf1/cip1 is increased, and GADD45α is increased.	[218]
		Range of concentrations assessed was 2.5–30 μmol/L.	Osteosarcoma MG63 osteosarcoma osteoblast cells (In vitro)	Akt is increased, and NF-κB is decreased.	[219]
		15 mg/kg/day for 15 days	Melanoma LiBr (In vivo)	Caspase-3 is decreased, and apoptosis is increased.	[220]
		10–300 μM for 48 h	Cell lines HCT-116, and SW-480	Cancer cell growth inhibition.	[103]
EGCG	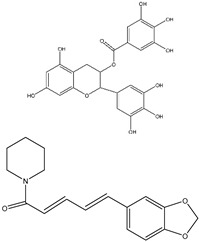	10, 20, 30, 40, 50, and 60 μg/mL	HT-29 cells for 24, 48, or 72 h	Reduces cell proliferation significantly by 61% after 48 h of treatment with 50 μg/mL of EGCG, relative to untreated.	[221]
		5 Μm EGCG with PDE3A inhibitor trequinsin	CSCs (Human)	Reduces the protein levels of FOXO3 and CD44 and upregulated cGMP expression, resulting in a significant reduction in the ability of CSCs to form colonies and spheroids.	[105]
		EGCG + drug IC50 (Cisplatin) (μM) 3.48 ± 0.24	OVCAR3 cell line	EGCG enhances cisplatin sensitivity by regulating the expression of the copper and cisplatin influx transporter CTR1 in ovary cancer.	[105]
		EGCG + drug IC50 (Cisplatin) (μM) 6.14 ± 0.38	SKOV3 cell line	EGCG enhances cisplatin sensitivity by regulating the expression of the copper and cisplatin influx transporter CTR1 in ovary cancer.	[105]
		EGCG + drug IC50 (5-FU) (μM) 5 ± 0.36	HCT-116 cell line	(EGCG) enhances the sensitivity of colorectal cancer cells to 5-FU by inhibiting GRP78/NF-kappaB/miR-155-5p/MDR1 pathway.	[105]
		EGCG + drug IC50 (5-FU) 11 ± 0.96	DLD1 cell line	(EGCG) enhances the sensitivity of colorectal cancer cells to 5-FU by inhibiting GRP78/NF-kappaB/miR-155-5p/MDR1 pathway.	[105]
		EGCG + drug IC50 (Doxorubicin) 0.66 ± 0.14 (μM)	BEL-7404 Cell line	Green tea catechins augment the antitumor activity of doxorubicin in an in vivo mouse model for chemoresistant liver cancer.	[105]
		EGCG + drug IC50 (Doxorubicin) 3.28 ± 0.34 (μM)	BEL-7404/DOX cell line	Green tea catechins augment the antitumor activity of doxorubicin in an in vivo mouse model for chemoresistant liver cancer.	[105]
		EGCG + drug IC50 (Doxorubicin) 29.89 ± 6.09 (μM)	CEM/ADR 5000 cell line	Modulation of multidrug-resistant cancer cells by EGCG, tannic acid, and curcumin.	[105]
		EGCG + drug IC50 (Tamoxifen) 5.8 ± 0.4 (μM)	MCF-7/TAM cell line	Combination of siRNA-directed gene silencing with epigallocatechin-3-gallate (EGCG) reverses drug resistance in human breast cancer cells.	[105]
		80 µM	T47D breast cancer cells	A significant decrease in hTERT gene expression causing apoptosis was observed.	[222]
		100 µM	HEC-18, HEC-18T, HEN-18, HEN-18S	Growth inhibition greater than 90% and induction of apoptosis was observed in HEC-18 and HEN-18. Telomerase was inhibited in all 4 cells.	[223]
		100, 200 µg/mL	Nasopharyngeal carcinoma cell line CNE2	Prevents CNE2 cells from proliferating, causes cell cycle block, apoptosis of the cells was promoted, and downregulation of the mRNA and protein expression of hTERT as well as c-Myc protein.	[224]
		EGCG: 100 µM RA: 1 µM	HeLa and TMCC-1	Combination treatment caused inhibition of telomerase, induction of apoptosis, and prevented cell proliferation.	[225]
		(50, 100 or 150 µM) of piperine for 72 h.	Breast cancer (Triple-negative breast cancer cells) (MDA-MB-231 and MDA-MB-468) in vitro	Inhibits the growth of p53-deficient cell lines by inhibiting the G1-S transition of the cell cycle and inhibiting CDK activity, leading to apoptosis.	[226]
Piperine	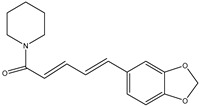	(50–200 mM) of piperine for 24, 48, and 72 h respectively.	Prostate cancer (LNCaP, PC-3, DU-145 and 22RV1 PCa cells)	Activate the caspase-3-dependent apoptotic pathway by suppressing the activation of phosphorylated STAT-3.	[227]
		(8, 16, and 20 μM) of piperine.	Ovarian cancer (A2780 cells)	Increase in mitochondrial cytochrome c release, which increased caspase-9 and caspase-3 activity (intrinsic pathway).	[228]
		(75–150 µM) of piperine.	Rectal cancer [Human rectal adenocarcinoma cells (HRT-18)]	Inhibition of G0/G1 cell cycle progression and formation of ROS.	[229]
		serial concentrations of piperine (25, 50, 100, 200, and 400 μg/mL)	Lung cancer (A549 cells)	Cell arrest of the phase G2-M in the cell cycle because of elevated levels of P53.	[230]
		0, 9, 18 of piperine and 36 µM curcumin for 24 h	Leukemia (HL60)	Release of mitochondrial cytochrome-c, which further initiates caspase-9/3 mediated cell apoptosis.	[231]
Piperine and curcumin	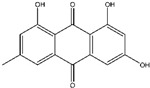	0–80 μM	Ovarian cancer A2780 and SK-OV-3	Inhibits cancer cells’ epithelial-mesenchymal transition by blocking the ILK/GSK-3β/slug signaling pathway.	[123]
Emodin	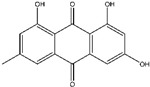 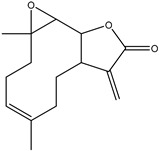	30 µmol/L	Cervical cancer SiHa, CaSki, and HaCaT	Increases caspase-3 activity and ROS production.	[116]
		25, 50, 100 μmol/L	Breast cancer MCF-7 EO771, 4T1, MCF7, and MDA-MB-231	Aryl hydrocarbon receptor (AhR) activators (agonists of AhR) increase the expression of CYP1A1. Suppressed TGF-β1 production.	[232] [119]
		0, 25, 50, and 100 μM	Kidney cancer 786-O and OS-RC-2	Increases ROS and the consequent activation of the JNK pathway. Increased phosphorylation levels of necroptosis-related proteins MLKL and RIP1.	[118]
		40 mM 25, 50 and 100 μmol/L	Hepatocellular carcinoma HepaRG SMMC-7721	Increase in BAX expression and a decrease in BCL-2 expression. Increased of cleaved caspase-3, -9, and PARP level. Induce the phosphorylation of ERK and p38. Inhibit the expression of p-c-Jun-N-terminal kinase (p-JNK). Suppress the activation of p-Akt.	[120] [123]
		15, 30, 60 μg/mL 60 μM	Colon cancer HCT116 CACO-2 CC-531	Increased PI3K, p-Akt, and VEGFR2 expressions. Inhibition of PI3K/Akt signalling pathway.	[122] [121]
		60 and 80 µmol/L	lung cancer A549 and H1299	Inhibition of ER stress with 4-PBA.	[124]
		20, 40 and 80 µmol/L	Bladder cancer T24 and T5637	Inhibits the expression of notch1.	[125]
		2.5–25 μM	Prostate cancer LNCaP	Caspase-3 activation.	[139]
Parthenolide	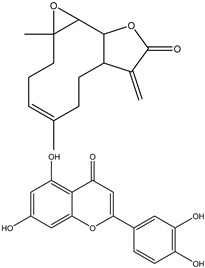	0–20 μM	Leukemia K562/ADM	Downregulation of NF-κB activity and mediated P-gp expression, increasing ROS, Bax/Bcl-2 ratio, and cytochrome C expression.	[140]
		50 μM	Breast cancer 231MFP and HCC38	Inhibits FAK1 activity and FAK1-dependent signaling pathway; activation of caspase-3/7.	[141]
		3–10 μM	Colorectal cancer HCT116, SW480, SW620, HT29, and Caco-2	Reduces the activity of USP7 and Wnt signaling, resulting in caspase-8 and-9 activation.	[142]
		UNK	Gastric cancer SGC7901	Downregulation of NF-κB activity and Bcl-2 expression, and upregulation of Caspase-8 activity.	[233]
		5, 10 and 20 μM	Squamous Cell Carcinoma Eca109 and KYSE-510	Reduces the levels of NF-κB, AP-1, and VEGF.	[143]
		4, 8 and 10 μM	Thyroid cancer TPC-1	Increase ROS levels and Bax expression; downregulation of Bcl-2.	[144]
		5–50 μM	Lung cancer A549 and H1299	Inhibits the IGF-1R Mediated PI3K/Akt/ FoxO3α signaling.	[145]
		4-8 μM	Renal cell carcinoma 786-O and ACHN	Inhibits the EMT, cancer stem cell markers, and the PI3K/Akt pathway.	[146]
		50, 100, and 200 µmol/L	Uveal melanoma C918 and SP6.5	Increases the expression of p21, Bax, caspase-3, and caspase-9; downregulation of Cyclin D1, Bcl-XL and Bcl-2.	[234]
		25 μM	p53-deficient cell lines	Reduces cell viability by inducing apoptosis in p53-deficient cell lines by significantly increasing the cell proportion at the sub-G0/G1-phase of the cell cycle and decreasing the cell proportion at S-phase.	[235]
Luteolin	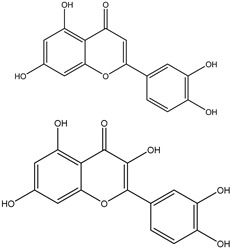	IC50 of 66.70 µmol/L (24 h) and 30.47 µmol/L (in 72 h)	colon cancerous LoVo cells	Cell cycle arrest at the G2/M phase that ultimately resulted in cellular apoptosis.	[235]
		in vitro with IC50 from about 3 to 50 μM	tumor cells	Suppress the proliferation of various kinds of tumor cells and inhibit tumor growth effectively in vivo when administered, e.g., in concentrations of 50 to 200 ppm in food.	[236]
		(10 mg/kg)	Sprague-Dawley rats	Inhibits the development of large tumors and can significantly lower the levels of vascular endothelial growth factor (VEGF).	[236]
		15 μM, 24 h	BxPC-3 human pancreatic cancer cells	Reduces nuclear GSK-3β and NF-κB p65 expression.	[237]
		30 μM	prostate carcinoma LNCaP cells	Induces cell apoptosis, up-regulating prostate-derived Ets factor (PDEF), and down-regulates androgen receptor (AR) gene expression.	[237]
		1–50 μM	Neuro-2a mouse neuroblastoma cells	Reduces cell viability by ~ 50%. Because the cytotoxic effect of luteolin was only modestly increased at the higher concentration (50 μM) of luteolin.	[237]
		6.25 to 100 µM	AGS cell line	Decreases the cell viability in a concentration-dependent way; 50% growth has been inhibited at a dose of 50 µM. When combined with SN-38, it enhances the anti-proliferation effect of the compound and enhances apoptosis, working synergistically with SN-38 to modulate GSK-3β/β-catenin signaling.	[238]
Quercetin	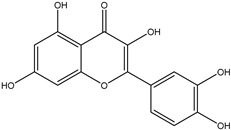	20 mg/kg BW	BALB/c nude mice injected with AGS cells	The administration of QUE alone (thrice weekly) or in conjunction with irinotecan (10 mg/kg once weekly) has resulted in a noteworthy decrease in tumor size by day 28. Additionally, tumor VEGF-R and VEGF-A levels, protein levels, and COX-2 gene expression have all decreased. Furthermore, this combination has reduced the TEM population.	[238]
		0 to 100 µM	MCF-7 and MDA-MB-231 cell lines	IC50 = 30 µM has decreased viability, elevated autophagy, suppressed migration rate, and decreased levels of MMP-2, MMP-9, and VEGF proteins. Moreover, decreased GLUT1, PKM2, LDHA, and glucose uptake have been all observed, as well as lactate production. In a similar vein, QUE has inhibited mTOR, p70-S6K, and Akt activation.	[239]
		50 mg/kg BW	Mice injected with MCF-7 cells	Prevents the progression of breast cancer and the metastasis of tumors. In tumor tissue, it also reduces the levels of p-Akt, PKM2, and VEGF.	[239]
		1.78 to 100 µM	PC3 cells	suppressed survival in a way that is time- and dose-dependent. QUE has shown an increased level of Cyt c, casp 3, casp 8, Bax, Bcl-2, p21Cip1, p27Kip1, and p53 at a concentration of 40 µM. QUE has also enhanced MKsi’s apoptotic effect, raised casp 3, and lowered the expression of the Survivin gene. Moreover, QUE has reduced the number of cells in the S-phase and encouraged the arrest of G1 phase cells. QUE has also reduced PI3K, Akt, and ERK1/2 phosphorylation and elevated PTEN expression while lowering p38, NFκB, and Survivin protein levels. When paired with MKsi, QUE has shown an outperformance in ERK1/2, p38, NFκB, and Survivin.	[240]
		150 µM at 72 h	B164A5 murine melanoma cells	Decreasing ECAR and OCR.	[241]
		12 to 100 µM	BC3, BCBL1, and BC1 PEL cell lines.	In a dose-dependent manner, QUE for 24 h has shown a decrease in cell growth and survival while having no effect on healthy B lymphocytes. The apoptotic rate has been elevated by QUE 50 µM, leading to an increase in the G1 cell phase, PARP cleavage, and nuclear fragmentation/condensation. QUE has also promoted the degradation of β-catenin and inhibited Aktser473 and mTOR at this concentration.	[164]
		0.6 to 300 µM	MCF-7, MDA-MB-231, HBL100 and BT549 breast cancer cells, and OVCAR5, TOV112D, OVCAR3, CAOV3 ovarian cancer cells.	Decreasing in concentration-dependent cell proliferation.	[242]
		50 µM for 24 h	HBL100 cells	Increasing the lactate depletion in the culture media and increasing the intracellular glucose accumulation.	[242]
		300 µM for 48 h	MCF-7 cells	Apoptosis has increased by 25%.	[242]
		0 to 200µM	HCT-15 and RKO cells	Promoting apoptosis and inhibiting cell viability and proliferation in cancer cells in a concentration-dependent manner, but not in normal cells.	[243]
		142.7 µM (IC50)	HCT-15 cells	Decreasing the production of lactate and consumption of glucose following a 4 h incubation. By increasing sensitization to 5-FU, QUE has enhanced the inhibitory effects of the compound on glucose metabolism.	[243]
		121.9 µM (IC50)	RKO cells	QUE has also improved the inhibition of glucose metabolism by increasing sensitization to 5-FU.	[243]
		25 to 75 mg/kg BW	DL mice	Reducing LDH-A activity, mRNA expression, and cell viability in a dose-dependent way without causing hepatic toxicity. Additionally, QUE has increased p53 mRNA expression while downregulating Akt gene/protein expression and p85a phosphorylation.	[244]
		26.5 µM	Ehrlich ascites tumor cells	Reducing the synthesis of lactate by 78% and Na+-K+-ATPase by 85%.	[245]
		13.25 to 66.17 µM	Ehrlich ascites tumor cells	After 10 min of treatment and at a concentration of 26.5 µM, QUE has inhibited Na+-K+-ATPase activity by 50%. In a concentration-dependent manner, QUE has inhibited both oxidative phosphorylation and aerobic glycolysis.	[246,247]
		33.09 µM	Ascites tumor cells	Inhibiting glycolysis and protein synthesis.	[248]
		25 µM	Rat thymocytes	Preventing the uptake of glucose brought on by a mitogenic stimulus.	[249]
		0.1 µg/mg protein	Ascites tumor cells	Lactate efflux has been 50% inhibited, resulting in an increase in the internal lactate concentration while decreasing intracellular pH.	[250]
		5 to 40 µM	HL60 cells	QUE and 2-DG together have been the main cause for the reduction in mitochondrial membrane potential, an increase in mIMP, caspase-dependent late apoptosis, and attenuated phosphorylation of Akt and rpS6. The rate of apoptosis is increased when PI3K/Akt phosphorylation inhibitors are used in conjunction with QUE (10 µM) and 2-DG (2 mM) at low concentrations.	[251]
		9.24 µM	Recombinant human PKM2 enzyme	50% less PKM2 activity has been mainly observed.	[252]
		Anthocyanins extracts (400 µg/mL)	A549 (lung cancer cells)	Inhibiting the invasion and migration of cancer (MMP-2 and MMP-9) and preventing it from proliferating (COX-2, C-myc, and cyclin D1).	[253]
Anthocyanins		Anthocyanidins, cyanidin, malvidin, peonidin, petunidin, and delphinidin (12.5–100 µM)	A549, H1299 (lung cancer cells)	Inducing apoptosis (Bcl2, PARP), inhibited the WNT and oncogenic Notch pathways (VEGF, pERK, cyclin D1, cyclin B1, b-catenin, c-myc, and MMP9).	[254]
	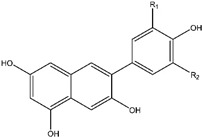	Anthocyanins extracts (40–320 µg GAE/mL, equivalent to 10.8–86.2 µg C3G/mL)	BT4747, MDA-MB-453, MDA-MB-231(Breast cancer cells)	The inhibition of proliferation and apoptosis by downregulating the Akt/mTOR and pro-survival Sirt1/survivin pathways, as well as a decrease in invasive/metastatic markers Sp1, Sp4, and VCAM-1.	[255]
	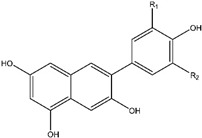	Anthocyanins extracts, delphinidin, cyanidin (100–600 mg/mL)	HCT-116, HT-29 (colon cancer cells), in silico analysis	Decreasing the levels of PD-1, PD-L1, and PVGF expression.	[256]
		Anthocyanin extracts (5 mg/mL; 5 mg/kg)	colon cancer stem cells; AOM induced carcinogen in A/J mice	Apoptosis induction and down-regulation of Wnt/b-catenin signaling.	[257]
		Anthocyanins extracts (400 µg/mL)	SNU-1, SNU-16 (gastric cancer cell)	Apoptosis induction (Caspase-3, Akt, Bax, XIAP).	[253]
		Anthocyanins extracts (0.2 mg/mL	PLC/PRF/5, HepG2, McArdle (liver cancer cell)	Induced autophagy (eIF2a, mTOR, Bcl-2) and apoptosis (Bax, cytochrome C, caspase-3).	[258]
		Anthocyanins (60–120 mM)	DU-145 (prostate cancer cell), tumor xenograft in nude mice	Inhibiting the androgen signaling (AR) and triggering apoptosis (p53, Bax, and Bcl 2).	[259]
		Anthocyanins extracts (100–200 µg/mL)	SK-Hep1 (liver cancer cell)	Reducing migration, decreasing adhesion, and suppressing proliferation to prevent metastasis (MMP-2, MMP-9).	[260]

## Data Availability

The datasets used and analyzed during the current study are available from the corresponding author upon reasonable request.

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
