# Peer review of "Natural Products and Altered Metabolism in Cancer: Therapeutic Targets and Mechanisms of Action"

_ijms, 2024, doi:10.3390/ijms25179593_

Round 1

Reviewer 1 Report

Comments and Suggestions for Authors

Dear Authors

This is an interesting manuscript about natural products and metabolism in cancer. 

It is an extensive review regarding mechanisms of action of different natural products and cancer.

All figures need to be improved their quality for publication.

Comments on the Quality of English Language

The English is fine. Some minor errors 

Author Response

Thank you for your comments. The manuscript was revised and corrected based on your comments. Attached is a detailed answers for you comments.

We hope that the revised manuscript will meet your expectations.

Best Regards,

Prof. Wamidh Talib

Reviewer 2 Report

Comments and Suggestions for Authors

Author Response

(The authors gave the same response as above.)

Reviewer 3 Report

Comments and Suggestions for Authors

The revision is limited just in few phenolic derivatives compounds, additionally you mention that this review is focus on recent articles to investigate the impact of natural products on altered metabolism in cancer, however not all are recent (See table 1). You should consider a limit of years in the review and mention in the title and consider to change it. The references are not in the format required for the journal. I include a pdf with additional observations.

1. References are not in the format stablished by the journal.

2. RV, RSV and RES appears in the description of Resveratrol, are them mean the same? Could you use the same acronym for them?

3. Pay attention in cursives words in whole manuscript.

4. In Line 280 you start with a reference, is this right??

5. Could you mention the figures in the text? On the other side sentences on the figures should be descriptive to explain by themself.

6. A revision on the redaction of EGCG should be done.

7. Since almost all reviewed compounds are phenolic derivatives, why did you decide include parthenolide?  

8. Could you improve the table 1?

9. Could you include the structures with best quality?

10. Could you revise the anthocyanin structure? It is right?

Author Response

(The authors gave the same response as above.)

Reviewer 4 Report

Comments and Suggestions for Authors

Despite the fact that the review entails a subject falls into the scope of the journal, it lacks the recent bibliography review and the organization of the review need extensive revision.

Comments on the Quality of English Language

The English quality can be further improved

Author Response

(The authors gave the same response as above.)
